# A systematic bi-genomic split-GFP assay illuminates the mitochondrial matrix proteome and protein targeting routes

Yury S Bykov[1]*[†], Solene Zuttion[2], Dunya Edilbi[1], Marina Polozova[3], Johanna Arnold[3], Sergey Malitsky[4], Maxim Itkin[4], Bruno Senger[2], Ofir Klein[1], Yeynit Asraf[1], Hadar Meyer[1], Hubert D Becker[2], Roza Kucharczyk[5], Maya Schuldiner[1]*

[1]Department of Molecular Genetics, Weizmann Institute of Science, Rehovot, Israel; [2]Université de Strasbourg, CNRS, Génétique Moléculaire, Génomique, Microbiologie, Strasbourg, France; [3]Department of Biology, Rheinland-Pfälzische Technische Universität (RPTU), Landau Kaiserslautern, Germany; [4]Metabolomics Unit, Weizmann Institute of Science, Rehovot, Israel; [5]Institute of Biochemistry and Biophysics, Polish Academy of Sciences, Warsaw, Poland

*For correspondence:
yury.bykov@rptu.de (YSB);
maya.schuldiner@weizmann.ac.il (MS)

Present address: [†]Quantitative Cell Biology, Rheinland-Pfälzische Technische Universität (RPTU), Kaiserslautern, Germany

## eLife Assessment

This study represents a **valuable** addition to the catalog of mitochondrial proteins. With the use of methodology based on the bi-genomic split-GFP technology, the authors generate **convincing** data, including dually localized proteins and topological information, under various growth conditions in yeast. The study represents a key basis for further functional and/or mechanistic studies on mitochondrial protein biogenesis.

**Abstract** The majority of mitochondrial proteins are encoded in the nuclear genome. Many of them lack clear targeting signals. Therefore, what constitutes the entire mitochondrial proteome is still unclear. We here build on our previously developed bi-genomic (BiG) split-GFP assay (Bader et al., 2020) to solidify the list of matrix and inner membrane mitochondrial proteins. The assay relies on one fragment ($GFP_{1-10}$) encoded in the mitochondrial DNA enabling specific visualization of only the proteins tagged with a smaller fragment, $GFP_{11}$, and localized to the mitochondrial matrix or the inner membrane. We used the SWAp-Tag (SWAT) strategy to tag every protein with $GFP_{11}$ and mated them with the BiG GFP strain. Imaging the collection in six different conditions allowed us to visualize almost 400 mitochondrial proteins, 50 of which were never visualized in mitochondria before, and many are poorly studied dually localized proteins. We use structure-function analysis to characterize the dually localized protein Gpp1, revealing an upstream start codon that generates a mitochondrial targeting signal and explore its unique function. We also show how this data can be applied to study mitochondrial inner membrane protein topology and sorting. This work brings us closer to finalizing the mitochondrial proteome and the freely distributed library of $GFP_{11}$-tagged strains will be a useful resource to study protein localization, biogenesis, and interactions.

## Introduction

The cell is an intricate molecular device. Full understanding of its functions requires a complete inventory of all the components tracing their positions and connections to each other. Of all eukaryotic

organisms, *Saccharomyces cerevisiae*, or budding yeast, comes closest to having such an understanding thanks to its simple organization and compact genome allowing for versatile manipulations (*Michaelis et al., 2023*). A systematic collection of strains where every gene in the genome is fused to a green fluorescent protein (GFP) at the C-terminus was one of the first genetic instruments to offer a glimpse into the global distribution of yeast proteins across all organelles and its plasticity in different conditions (*Huh et al., 2003*; *Breker et al., 2013*). A new generation of SWAp-Tag (SWAT) libraries that allow quick substitution of the N-terminal or C-terminal tag with any DNA sequence of choice greatly expanded protein visualization possibilities (*Yofe et al., 2016*). The use of brighter fluorophores and N-terminal tags combined with stronger promoters helped to gain more insight into intracellular protein distribution (*Meurer et al., 2018*; *Weill et al., 2018*; *Dubreuil et al., 2019*). Despite all these advances and systematic studies, many proteins are not yet fully characterized in terms of their subcellular and sub-organellar distribution.

The mitochondrial proteome is one of the most challenging to precisely map. First, many of its ~800 proteins are dually localized and also have a certain fraction in an additional compartment (so-called 'eclipsed' distribution *Regev-Rudzki and Pines, 2007*). Such eclipsed proteins are often not clearly distinguished as mitochondrial, especially if only a minority of the protein is in mitochondria and its signal is masked by high cytosolic levels. In addition, mitochondrial proteins are distributed to different sub-locations inside the organelle: outer membrane (OM), intermembrane space (IMS), inner membrane (IM), and matrix. Most of the proteins are produced in the cytosol and are imported based on the targeting signals encoded in their amino acid sequence (*Herrmann and Bykov, 2023*). On the OM, these signals are recognized and threaded through by the translocase of the outer membrane (TOM) complex. To cross the IM, the imported proteins bind the translocase of the IM (TIM23) complex. Metabolite transporters get inserted into the IM by the TIM22 complex without being imported into the matrix first. Importing soluble matrix proteins requires engagement of the import motor that binds the TIM23 complex from the matrix side and uses the energy of ATP hydrolysis to aid translocation. The IM is a major protein delivery destination in mitochondria. Transmembrane domains (TMDs) can be either laterally sorted into the IM after translocation arrest in the TIM23 complex (stop-transfer mechanism), or completely imported into the matrix and inserted by the Oxa1 translocase via the so-called conservative sorting pathway (*Stiller et al., 2016*). Both stop-transfer and conservative sorting pathways can act on different TMDs within one protein (*Bohnert et al., 2010*; *Park et al., 2013*). The mechanisms and machineries of mitochondrial protein import and membrane insertion are well-studied thanks to experiments on isolated mitochondria; however, the substrate repertoire of each pathway is still not fully mapped.

Since only a few artificially synthesized model substrates were used in in vitro sorting experiments, there is little understanding of how the mitochondrial protein import system handles hundreds of different proteins that are simultaneously imported in living cells (*Bykov et al., 2020*). For example, we have recently discovered that many mitochondrial proteins do not have conventional, well-predicted, targeting signals, but nevertheless are efficiently translocated into mitochondria (*Bykov et al., 2022*). Conversely, there are a lot of cytosolic proteins with high mitochondrial targeting signal prediction scores that are not imported (*Woellhaf et al., 2014*; *Mark et al., 2023*). It is also not clear how the cell precisely directs the proteins to sub-mitochondrial locations. To study these mechanisms in vivo, it is essential to first obtain a detailed inventory of the mitochondrial proteome, information on the sub-mitochondrial localization, and the topology of each protein.

High-throughput microscopy of GFP-tagged yeast strain collections enabled a great basis for mapping the mitochondrial proteome in living cells. However, this technique has been limited by its inability to show sub-mitochondrial location and topology. Another limitation is that proteins with eclipsed distribution and only a small mitochondria-localized fraction (echoform) are often missed, while the proportion of the fraction might vary with growth conditions (*Regev-Rudzki and Pines, 2007*). Biochemical fractionation of the organelles and mitochondrial sub-compartments, followed by mass-spectrometry-based proteomics can overcome these limitations. The most comprehensive mitochondrial protein distribution and topology inventories were produced using these approaches (*Vögtle et al., 2017*; *Morgenstern et al., 2017*; *Di Bartolomeo et al., 2020*). However, they are limited by the inability to truly purify the organelle or its sub-compartments leading to false positives and negatives.

Split reporters are another powerful tool to detect protein localization in living cells with high sensitivity. A recent study that used a nuclear-encoded split $\beta$-galactosidase reporter indeed uncovered many new mitochondrial proteins (*Mark et al., 2023*).

In this work, we aimed to combine the advantages of high-throughput microscopy that offers quick and sensitive protein visualization in living cells and the precision of split reporters to highlight individual mitochondrial sub-compartments. We made use of a recently developed bi-genomic split-GFP assay (BiG Mito-Split-GFP, hereafter shortened as BiG Mito-Split) where a larger fragment of GFP, $GFP_{1-10}$ is encoded in the mitochondrial DNA, and the small fragment $GFP_{11}$ is fused with a nuclear-encoded protein (*Bader et al., 2020*). In this way, only the matrix-localized fraction of the $GFP_{11}$-tagged protein produces a fluorescence signal. The advantage of the BiG Mito-Split assay over the previously used split reporters is that matrix-limited expression of $GFP_{1-10}$ completely excludes the possibility of cytosolic interaction of $GFP_{1-10}$ and $GFP_{11}$ before import, an artifact that is hard to completely avoid when both fragments are encoded in the nuclear genome. To take this approach to the whole-proteome level, we used the SWAT method and a C' SWAT acceptor library (*Yofe et al., 2016*; *Meurer et al., 2018*) to produce a whole-genomic library of C' tagged $GFP_{11}$ proteins and after mating them with a strain expressing the mitochondrial $GFP_{1-10}$, tested each of them for the presence of a matrix localized fraction with high-throughput microscopy. This approach allowed us to directly visualize many mitochondrial proteins that were not accessible by microscopy before. By combining our data with targeting signal predictions, we show that a very small fraction of cytosolic proteins with high mitochondrial targeting sequences (MTS) prediction scores can actually be imported into the matrix, highlighting a strict selectivity filter for soluble proteins. We also show an unexpectedly low selectivity of TMD-containing protein sorting at the IM. Overall, our studies uncover new mitochondrial proteins and, beyond that, the collection (which will be freely distributed) and dataset will be a useful resource to enable in vivo studies of mitochondrial protein targeting, translocation, and intra-mitochondrial sorting as well as more generally protein localization and interactions.

## Results
### Creation of a whole proteome GFP₁₁ library enables accurate and systematic visualization of the mitochondrial matrix-facing proteome

To perform the BiG Mito-Split assay on a whole proteome level, we first used the SWAT method to create a haploid collection of strains (library) where in each strain one gene is fused with $3×GFP_{11}$. We chose to use the C-SWAT library that allows C-terminal tagging (*Meurer et al., 2018*) since most mitochondrial proteins contain targeting signals at their N-termini. The original C-SWAT library contains ~5500 strains where each gene is fused to an acceptor cassette containing the *URA3* gene. Using automated mating and selection techniques, we introduced a donor cassette expressing the $3xGFP_{11}$ tag and $MTS_{Su9}$-mCherry as a mitochondrial marker. Recombination and negative selection for the loss of the *URA3* marker yielded a new haploid C'–$3×GFP_{11}$ collection (*Figure 1*).

To introduce mtDNA-encoded $GFP_{1-10}$, we depleted this collection of its own mtDNA and immediately mated it with the BiG Mito-Split strain, thereby producing diploids with restored respiratory capacity and both fragments of split-GFP encoded by nuclear and mitochondrial genomes (*Figure 1*).

The diploid BiG Mito-Split collection was imaged in six conditions representing various carbon sources and a diversity of stressors the cells can adapt to: logarithmic growth on glucose as a control carbon source and oleic acid as a poorly studied carbon source; post-diauxic (stationary) phase after growth on glucose where mitochondria are more active and inorganic phosphate ($P_i$) depletion that was recently described to enhance mitochondrial membrane potential (*Ouyang et al., 2024*); as stress conditions, we chose growth on glucose in the presence of 1 mM dithiothreitol (DTT) that might interfere with the disulfide relay system in the IMS, and nitrogen starvation as a condition that may boost biosynthetic functions of mitochondria. DTT and nitrogen starvation were earlier used for a screen with the regular C'-GFP collection (*Breker et al., 2013*). Another important consideration for selecting the conditions was the technical feasibility to implement them on automated screening setups. The imaging and initial visual analysis revealed 543 strains that showed a putative signal in at least one condition (*Supplementary file 1*).

We selected these strains for a more detailed study and fluorescence signal quantification to verify the initial result of the visual analysis. The experiment was performed in two 384-well plates. We had

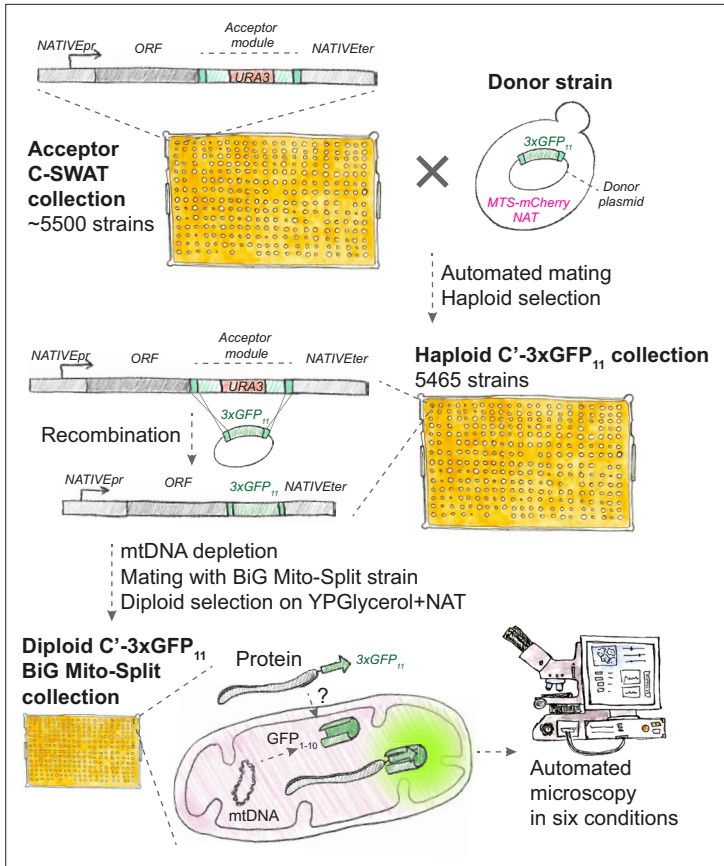

**Figure 1.** Schematic of the BiG Mito-Split collection creation and microscopy screening. To create a whole-proteome fusion library, the acceptor C-SWAT library where each open reading frame (ORF) under its native promoter (NATIVEpr) is tagged with a cassette (acceptor module) that contains *URA3* marker and homology linkers (dark green) was crossed with a donor strain that carries a plasmid with the desired tag (*3×GFP₁₁*) and genomically integrated mitochondrial marker $MTS_{Su9}$-mCherry and a NAT selection marker. Following sporulation and haploid selection, recombination is initiated, and the acceptor module is swapped with the donor tag, following negative selection for the loss of *URA3* marker. To assay mitochondrial localization, the resulting haploid C'- collection is deprived of its mtDNA on ethidium bromide and crossed with the BiG Mito-Split strain carrying $GFP_{1-10}$ in its mtDNA, diploids selected on respiratory media supplemented with NAT. The diploid collection is imaged using an automated fluorescence microscope and only proteins with their C-termini localized in the matrix complement split-GFP and can be detected.

the possibility to additionally include in the quantitation 123 non-fluorescent control strains. We also added 102 strains of proteins that were previously reported to be localized to the matrix or IM in proteomics studies (*Vögtle et al., 2017*; *Morgenstern et al., 2017*) and that we may have potentially overlooked in the visual analysis of whole-genomic screens (see *Supplementary file 1* legend). To compare raw fluorescent measurements across different plates and conditions, we normalized them to the average and standard deviation observed for multiple non-fluorescent controls present on each plate (*Figure 2—figure supplement 1A*, see Materials and methods) (The full dataset is provided in *Supplementary file 1*). Of note, out of 102 extra strains that were picked based on the previous annotations only three showed signal. This means that our initial imaging had a very low false negative rate and that our two-step screening was an effective strategy to identify matrix and inner membrane proteins that can be visualized with the BiG Mito-Split collection.

Despite the highly diverse conditions that we picked—constituting both stressed and metabolic changes—the normalized fluorescence measurements in different conditions correlated very strongly with each other (*Figure 2*). The differences were most likely defined by general upregulation or down-regulation of mitochondrial gene expression reflected in the linear regression slope. For example, the intensities are overall higher in stationary phase compared to logarithmic growth on glucose, as

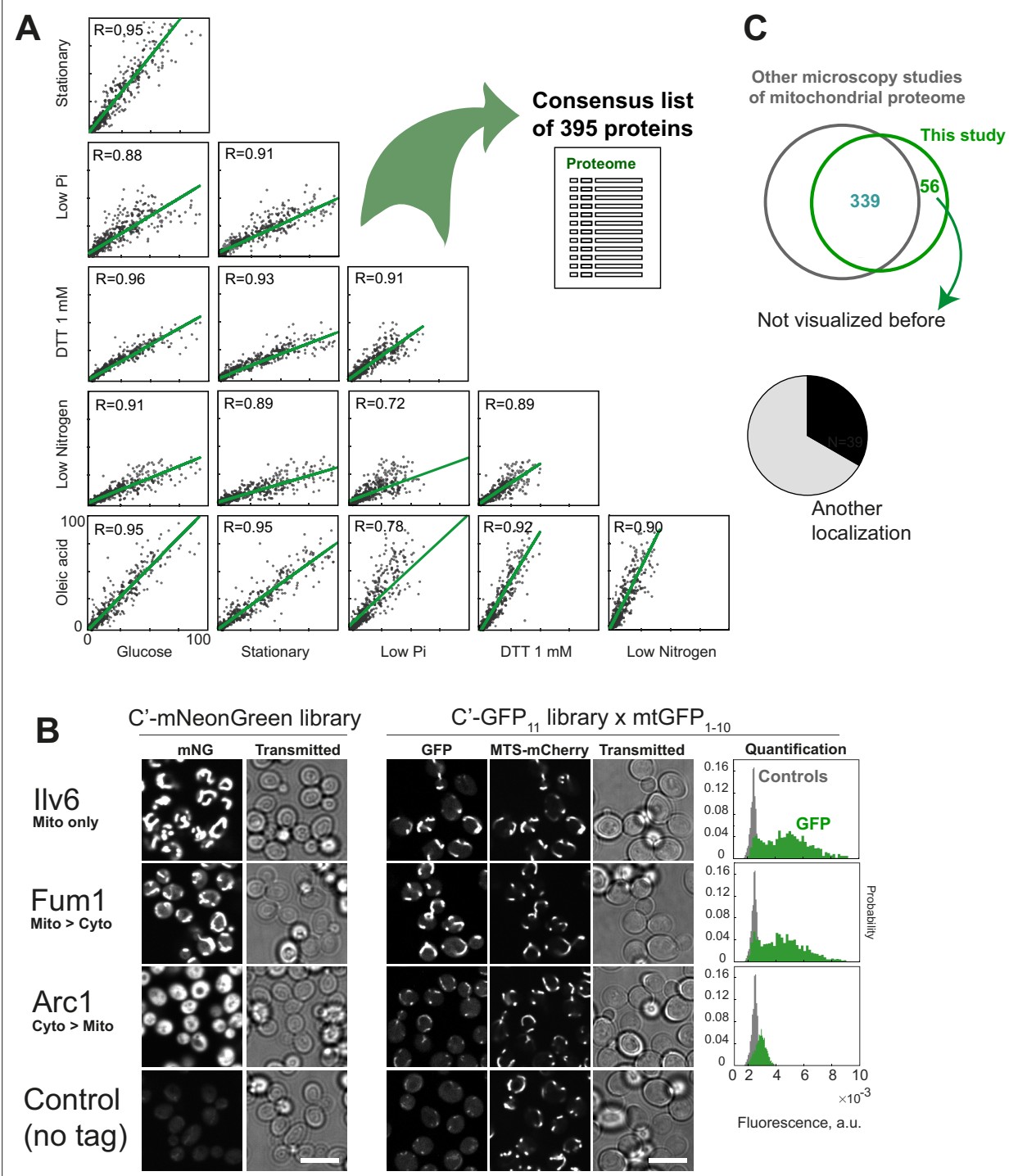

**Figure 2.** Visualization of mitochondrial proteins with the BiG Mito-Split collection. (**A**) Scatter plots of correlation between normalized fluorescence intensities of each strain imaged in different conditions, the line and value of linear regression are shown. (**B**) Examples of fluorescent micrographs of strictly mitochondrial and dually localized proteins visualized by full-length fluorescent protein tags (C-SWAT mNeonGreen (mNG) library, left) or using the BiG Mito-Split collection (right). Quantification of the GFP signal in mitochondria marked by MTS$_{Su9}$-mCherry in BiG Mito-Split strains relative to pooled controls without GFP$_{11}$ is shown in the far right. (**C**) Comparison of the proteins visualized in this study with the previous microscopy studies, and the breakdown of newly visualized proteins into the ones previously found in another location, or never studied with high-throughput microscopy before. Scale bars 10 µm.

The online version of this article includes the following source data and figure supplement(s) for figure 2:

**Source data 1.** The original micrographs of the mNeonGreen-tagged strains displayed in *Figure 2B* (left).

*Figure 2 continued on next page*

*Figure 2 continued*

**Source data 2.** The original micrographs of the 3×GFP11-tagged strains from the diploid BiG Mito-Split collection shown in *Figure 2B* (right).

**Figure supplement 1.** Quantitative analysis and normalization of GFP signal in different growth conditions.

expected, since the former condition supposedly displays higher respiration. Our dataset did not reveal large groups of proteins that conditionally become imported into mitochondria under certain metabolic states or stresses. Proteins that were only detected in a few conditions were mostly of low abundance (*Supplementary file 1*), so we could not conclude whether their expression or import are differentially regulated, or rather that we do not detect them in some conditions due to lower signal-to-noise ratios.

To compare our findings with published data, we created a unified list of 395 proteins that are observed with high confidence using our assay indicating that their C-terminus is positioned in the matrix (*Figure 2—figure supplement 1B–D*, *Supplementary file 1*). Most proteins we identified are well-studied, exclusive mitochondrial matrix or IM proteins that were already reported by previous microscopy studies. Such proteins appear the same when tagged with a full-length fluorescent protein (FP) and when visualized with BiG Mito-Split showing strong mitochondrial signal and no signal in the cytosol (Ilv6, *Figure 2B*). We also found known dually localized proteins like Fum1 whose mitochondrial fraction is clearly visualized by full-length FP fusion while the other localization (cytosol) is also clearly visible with the full-length FP (Fum1, *Figure 2B*). Finally, for some proteins mitochondrial localization was impossible to assign based on previously studied full-length FP fusions and they were for the first time visualized as mitochondrial in our dataset (Arc1, *Figure 2B*). Overall, compared to all the previous microscopy studies using yeast collections (*Huh et al., 2003*; *Breker et al., 2013*; *Weill et al., 2018*; *Meurer et al., 2018*; *Dubreuil et al., 2019*), we visualized 56 new proteins (*Supplementary file 1*). Out of those, 39 were visualized in other locations meaning that in this study, we managed to highlight eclipsed mitochondrial echoforms. There were also 17 proteins never before visualized by

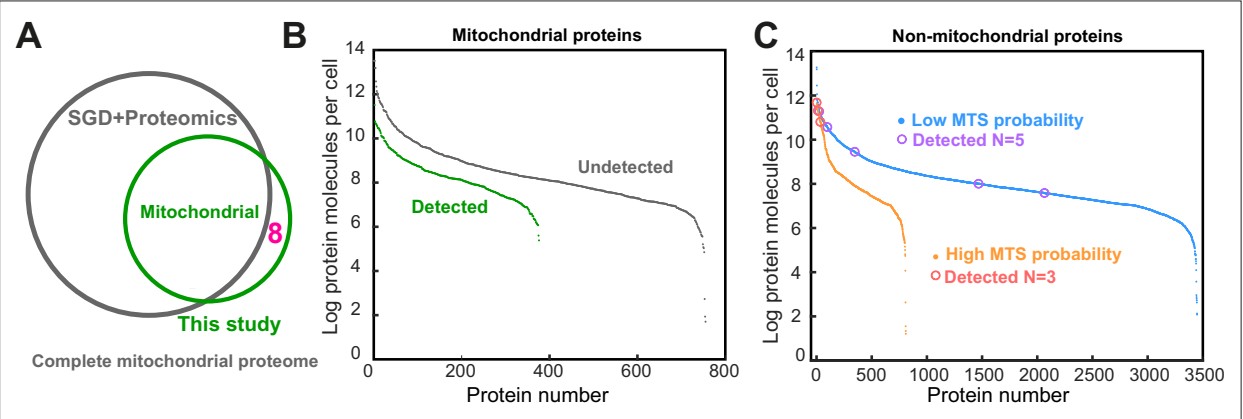

**Figure 3.** Comparison of the proteins visualized in the BiG Mito-Split collection with previous studies of the mitochondrial proteome. (**A**) A Venn diagram showing a comparison of proteins defined from this work with manual annotations in the *Saccharomyces* Genome Database (SGD) and high-throughput proteomics reveals eight proteins previously not associated with mitochondria. (**B**) Mitochondrial proteins visualized in our screen (green) compared to all other mitochondrial proteins (gray, list compiled as in panel **A**) ranked by unified protein abundance (*Ho et al., 2018*). (**C**) Non-mitochondrial proteins ranked by unified protein abundance and split into a group with a high mitochondrial targeting signal prediction score (>0.7) or a low score (according to *Monteuuis et al., 2019*) with the eight non-mitochondrial proteins visualized in this work (panel **A**) marked with circles. Only the proteins with known unified abundance were analyzed (*Ho et al., 2018*).

The online version of this article includes the following source data and figure supplement(s) for figure 3:

**Source data 1.** The spreadsheets and the MATLAB script used to produce the graphs displayed in *Figure 3*.

**Figure supplement 1.** Proteins observed in this study compared to other works.

**Figure supplement 2.** The influence of protein essentiality and complex assembly on the assay performance, and the verification of selected strains.

**Figure supplement 2—source data 1.** PDF file containing the original western blots for *Figure 3—figure supplement 2B*.

**Figure supplement 2—source data 2.** Original files of western blot analysis displayed in *Figure 3—figure supplement 2B*.

**Figure supplement 3.** The similarity between r-proteins and mitochondrial proteins with cleavable presequences.

microscopy in high-throughput studies (*Figure 2C*). Thus, our work significantly expands the microscopic toolkit for mitochondrial protein visualization.

## BiG Mito-Split sensitivity and the definition of mitochondrial proteome

Next, we aimed to determine whether we can add new proteins to the mitochondrial proteome. For this, we compared our findings with data obtained by multiple alternate methods. As the most reliable sources, we used manual annotations from the *Saccharomyces* Genome Database (SGD, https://yeastgenome.org/; *Wong et al., 2023*) that extensively cover individual protein localization deduced using such methods as fractionation, western blotting, or manual fluorescence microscopy. From high-throughput annotations, we used the two most confident proteomics datasets (*Vögtle et al., 2017*; *Morgenstern et al., 2017*). Compared to this unified list, we still found eight proteins never reported to be mitochondrial (*Figure 3A*, *Supplementary file 1*).

It is harder to estimate how many proteins were missed in our assay. We expect to visualize all soluble matrix proteins and the IM proteins with their C-termini facing the matrix. There is no reliable annotated dataset on sub-mitochondrial protein distribution except the two mentioned proteomic studies, and only one accessible IM protein topology annotation (*Morgenstern et al., 2017*). When compared to the list of 194 soluble matrix proteins (*Vögtle et al., 2017*), we identify 136 (70%) which is superior to using a split-enzymatic system (*Figure 3—figure supplement 1A*; *Mark et al., 2023*). To check if the low abundance proteins were more likely to be overlooked by the BiG Mito-Split assay, we compared the abundances of detected and undetected proteins and found that many low abundance proteins still can be visualized (*Figure 3B*). Comparison with the high confidence mitochondrial proteome quantification (*Morgenstern et al., 2017*) showed a good agreement of fluorescence signal strength and protein abundance. It also revealed that some proteins might have preferred interaction with newly synthesized mtGFP$_{1-10}$ but did not show any specific functional protein groups that could not be detected by our method (*Figure 3—figure supplement 1B*).

Next, we checked the two categories of proteins likely to give biased results in high-throughput screens of tagged collections: proteins essential for viability, and molecular complex subunits. To look at the first category, we split the proteomic dataset of soluble matrix proteins (*Vögtle et al., 2017*) into essential and non-essential ones according to the annotations in the SGD (*Wong et al., 2023*). We found that there was no significant difference in the proportion of detected proteins in both groups (17 and 20% accordingly), despite essential proteins being less represented in the initial library (*Figure 3—figure supplement 2A*). From the three essential proteins of the (*Vögtle et al., 2017*) dataset for which the strains present in our library but showed no signal, two were nucleoporins Nup57 and Nup116, and one was a genuine mitochondrial protein Ssc1. Polymerase chain reaction (PCR) and western blot verification showed that the Ssc1 strain was incorrect (*Figure 3—figure supplement 2B*). We conclude that essential proteins are more likely to be absent or improperly tagged in the original C'-SWAT collection, but the essentiality does not affect the results of the BiG Mito-Split assay.

To examine the influence of protein complex assembly on the performance of the BiG Mito-Split assay, we analyzed the published structures of the mitoribosome and ATP synthase (*Desai et al., 2017*; *Srivastava et al., 2018*; *Guo et al., 2017*) and classified all proteins as either having C-termini facing in, or out of the complex. There was no difference between the 'in' and 'out' groups in the percentage observed in the BiG Mito-Split collection (*Figure 3—figure supplement 2A*) suggesting that the majority of the GFP$_{11}$-tagged proteins have a chance to interact with GFP$_{1-10}$ before (or instead of) assembling into the complex. PCR and western blot verification of eight strains with the tagged complex subunits for which we observed no signal showed that mitoribosomal proteins were incorrectly tagged or not expressed, and the ATP synthase subunits Atp7, Atp19, and Atp20 were expressed (*Figure 3—figure supplement 2B*). Atp19 and Atp20 have their C-termini most likely oriented towards the IMS (*Guo et al., 2017*) while Atp7 is completely in the matrix and may be the one example of a subunit whose assembly into a complex prevents its detection by the BiG Mito-Split assay.

Based on the presence of many low abundance proteins and molecular complex components in our dataset, we conclude that the BiG Mito-Split assay can detect relatively low amounts of proteins translocated to the matrix, but some proteins can be missed regardless of their abundance. Many proteins are missing because their C-termini are facing the IMS. Hence, we only draw conclusions from the presence of signal and not from its absence.

We find eight proteins that were never connected to mitochondria before. They comprise five components of the cytosolic ribosome (r-proteins), one dubious open reading frame, the proteasome regulator Blm10, and a putative quinone oxidoreductase Ycr102c (*Supplementary file 1*, *Figure 3—figure supplement 2B*). The high proportion of r-proteins that are very unlikely to have a function in mitochondria made us wonder if the mitochondrial proteome determined before is essentially complete and these eight proteins represent protein import noise caused by very high similarity to mitochondrial targeting sequences. To visualize how many proteins in the cytosol may be mistaken for mitochondrial ones, we split all the non-mitochondrial proteome ranked by abundance into having high targeting signal prediction score and low score (*Figure 3C*). The eight newly found proteins belong to both groups, but the ones with high prediction scores are highly abundant. Five out of eight proteins are components of the cytosolic ribosome (r-proteins). In agreement with previous reports (*Woellhaf et al., 2014*), we find that their unique properties, such as charge, hydrophobicity, and amino acid content, are indeed more similar to mitochondrial proteins than to cytosolic ones (*Figure 3—figure supplement 3*). Additional experiments with heterologous protein expression and in vitro import will be required to confirm the mitochondrial import and targeting mechanisms of these eight non-mitochondrial proteins. The data highlights that out of hundreds of very abundant proteins with high prediction scores only a few are actually imported and highlights the importance of the mechanisms that help to avoid translocation of wrong proteins (*Oborská-Oplová et al., 2025*). Our dataset confirms that the selectivity filter recognizing targeting signals at the OM is very strict and BiG Mito-Split collection can be a valuable tool to study the mechanisms maintaining this selectivity.

## Visualization of poorly studied dually localized proteins

We next aimed to determine if our assay helped to confirm localizations of poorly studied proteins that might have potential new functions in mitochondria. We compared our 56 newly visualized proteins (*Figure 2C*, *Supplementary file 1*) with manual annotations and proteomic data and noticed that there are 20 proteins that were only found to be mitochondrial in high-throughput studies and not confirmed by other methods. We also uncovered a second localization for the abundant cytosolic glycerol-3-phosphate phosphatase Gpp1 that is in a reverse situation. It escaped detection in all proteomic studies and is not annotated to be mitochondrial in SGD. However, a work that aimed to find new mitochondrial proteins arising from alternative translation starts confirmed that Gpp1-Protein A fusion has a mitochondrial fraction (*Monteuuis et al., 2019*). A full-length FP fusion of Gpp1 gives only cytosolic signal (*Dubreuil et al., 2019*). Thus, we confirm that Gpp1 is a dually localized protein with eclipsed distribution.

We wondered if the 20 poorly studied proteins are also dually localized. The full-length FP fusion data is not available for them, and we relied on the whole-cell to mitochondria abundance ratio previously calculated (*Vögtle et al., 2017*). The log10 of this ratio is less than zero if the protein is predominantly mitochondrial and more than zero if the protein has a significant fraction localized elsewhere (*Vögtle et al., 2017*). Most of our newly visualized proteins are in the dual-localized protein region with the log10 of the ratio more than zero (*Figure 4A*). This confirms the utility of the BiG Mito-Split collection to uncover eclipsed proteins and to aid in deciphering their targeting pathways and functions.

It is important to study such eclipsed proteins because they can use diverse mechanisms to ensure the dual localization. Understanding those can help to shed light on the factors maintaining protein import fidelity. We aimed to get an idea of the mechanisms the newly visualized dually localized proteins can use. For this, we checked in the published dataset (*Monteuuis et al., 2019*) if any of those can have an alternative translation start site that gives rise to a targeting sequence. The 20 proteins we found and Gpp1 were split into three categories: no alternative start sites but consistently high targeting signal prediction; no alternative start with low prediction; and alternative start giving rise to an echoform with a strong signal prediction (*Figure 4B*).

We wondered if Arc1 that has a low MTS prediction score and no alternative start to produce an echoform with a targeting signal, indeed contains a targeting signal within its annotated amino-acid sequence. We expressed Arc1-GFP$_{11}$ and a well-studied mitochondrial protein Cha1-GFP$_{11}$ from a plasmid with a heterologous promoter and their canonical start codons in a haploid BiG Mito-Split strain (*Figure 4C*). Both proteins revealed a GFP signal when observed by confocal fluorescence

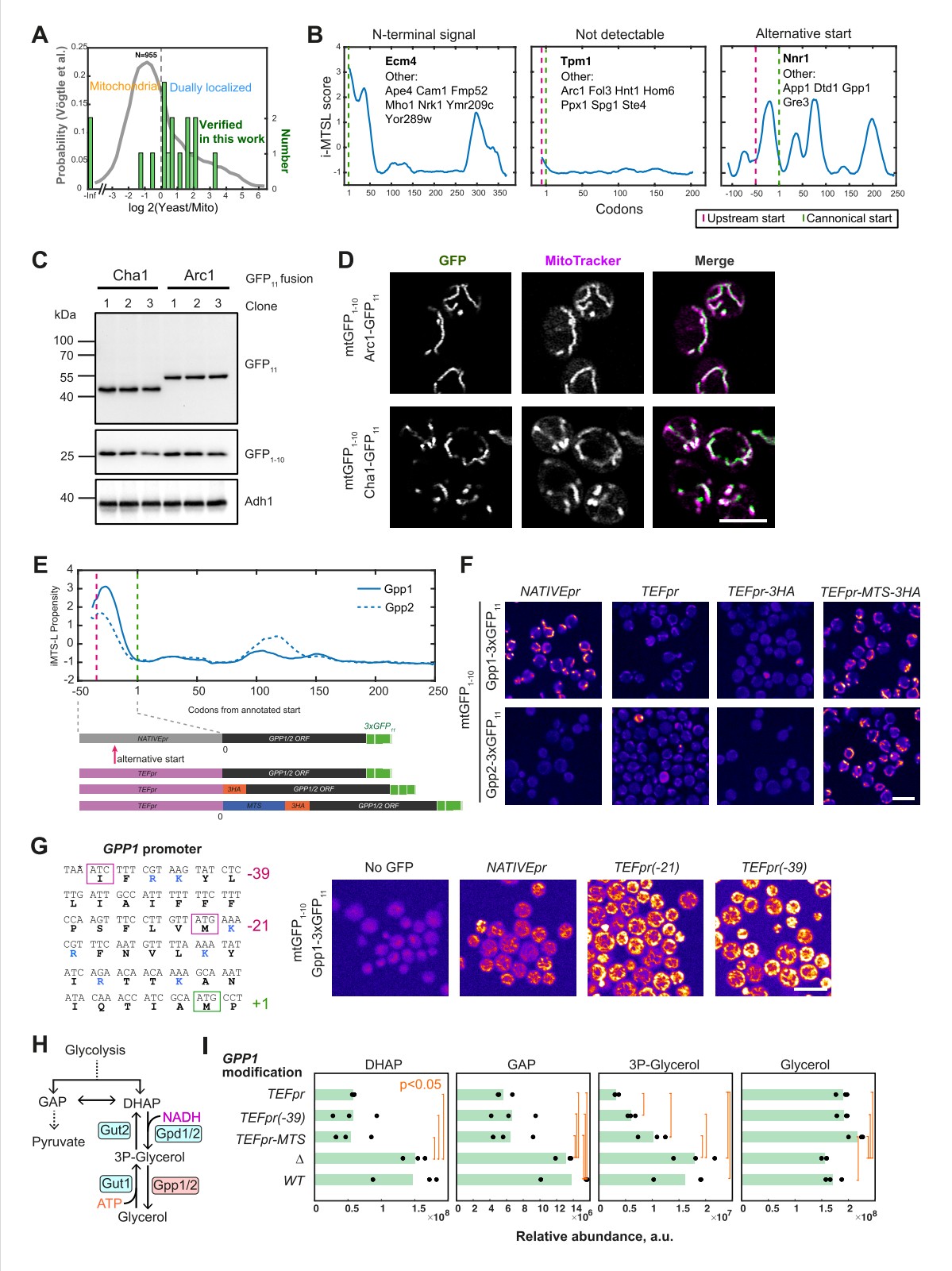

**Figure 4.** Dually localized proteins and their targeting signals. (**A**) Proteins visualized in this work and only found by high-throughput proteomics before tend to have high whole cell to mitochondria ratio (***Vögtle et al., 2017***) indicating that they are dually localized; only the proteins for which the ratio is known are plotted. (**B**) Three different types of potential dually localized proteins can be found: those predicted to have an N-terminal targeting signal and no alternative start codon, those with no targeting signal or alternative start prediction, those where an alternative start generates an echoform with

*Figure 4 continued*

high-scoring prediction; for each type, one example is shown with its graph of the i-MTSL-score, canonical start codon (green dashed line), alternative start codon (magenta dashed line), and other similar proteins listed on the plot. (**C**) Western blot verifying the expression of GFP₁₁-tagged Cha1 and Arc1 cloned into an expression plasmid with their canonical start codons under a heterologous promoter and transformed into a haploid BiG Mito-Split strain; three different clones from each transformation are shown; primary antibody used for decoration is shown on the right. (**D**) Confocal fluorescence microscopy of haploid BiG Mito-Split of the strains analyzed in (**C**), one clone is shown for each. (**E**) Gpp1 and Gpp2 i-MTSL start codons prediction (top), and schematics of generated constructs with mutated promoters and N-termini (bottom). (**F**) Fluorescence microscopy of the strains with mtDNA-encoded GFP₁₋₁₀ where Gpp1 and Gpp2 were tagged with 3×GFP₁₁ at the C-terminus (*NATIVEpr*) and then native promoter was substituted to *TEF2pr*, or *TEF2pr* followed by 3×HA tag, or MTS$_{Su9}$–3×HA. (**G**) Left: sequence of the *GPP1* promoter region with upstream on the canonical start codon (ATG, +1) showing the most upstream non-canonical start (ATC, –39) and an additional ATG (–21); right: fluorescence micrographs of the cells without GFP and of a haploid BiG Mito-Split strain where genomically-tagged *GPP1–3×GFP₁₁* has a native promoter or a *TEF2pr* integrated before the two non-canonical starts shown to the left. (**H**) The scheme of glycerol synthesis via by Gpp1/2 from the glycolysis-derived compounds glyceraldehyde-3-phosphate (GAP), dihydroxyacetone-phosphate (DHAP), and glycerol-3-phosphate; the relevant enzymes are shown, some details on additional metabolites are omitted. (**I**) Quantification of glycerol synthesis pathway metabolites by mass spectrometry in the strains with N-terminally modified Gpp1 (*TEFpr*, *TEFpr*(–39), and *TEFpr-MTS*), Gpp1 deletion strain (Δ), and the control with wild-type (WT) Gpp1; each quantification was performed on N=3 biological replicates; the averages with Welch-corrected t-test p-values less than 0.05 are connected by orange brackets. Scale bars are 10 μm (**F, G**) and 5 μm (**D**).

The online version of this article includes the following source data for figure 4:

**Source data 1.** PDF file containing the original western blots for *Figure 4C*.

**Source data 2.** Original files of western blot analysis displayed in *Figure 4C*.

**Source data 3.** The original micrographs of the strains displayed in *Figure 4D*.

**Source data 4.** The original micrographs of the strains displayed in *Figure 4F*.

**Source data 5.** The original micrographs of the strains displayed in *Figure 4G*.

microscopy (*Figure 4D*). This means that Arc1 harbors some poorly predicted mitochondrial targeting signal for which its exact location and recognition mechanism is still to be uncovered.

Then we turned to confirm that Gpp1 indeed contains an alternative start giving rise to an MTS. Gpp1 fell in the third category and had one of the highest signals in our dataset being consistently present in the matrix in all studied conditions. In the cytosol, Gpp1 dephosphorylates glycerol-3-phosphate and produces glycerol. This reaction is important under anaerobic conditions when mass glycerol production from glyceraldehyde-phosphate is used as a sink of reducing equivalents (*Pahlman et al., 2001*). Gpp1 has a paralog Gpp2 that arose from the whole genome duplication. Gpp2 plays an overlapping role and is expressed at lower levels unless upregulated by osmotic stress. Both proteins were suggested to have alternative start sites (*Monteuuis et al., 2019*) that give rise to mitochondrial echoforms (*Figure 4E*). To prove this, we tagged Gpp1 and Gpp2 with 3×GFP₁₁ in a haploid BiG Mito-Split strain. Interestingly, we find that Gpp1-3×GFP₁₁ has a strong mitochondrial signal, but Gpp2-3×GFP₁₁ does not (*Figure 4F*). Then we eliminated the alternative start codons upstream of the canonical ATG by introducing either a *TEF2* promoter (*TEF2pr*) alone, *TEF2pr* followed by 3×HA tag or a *TEF2pr* followed by MTS$_{Su9}$–3×HA (*Figure 4F*). For both proteins, introduction of *TEF2pr* or *TEF2pr-3×HA* abolished mitochondrial localization, while *TEF2pr*- MTS$_{Su9}$–3×HA led to strong mitochondrial fluorescence (*Figure 4F*). To prove that the upstream sequence in *GPP1* promoter indeed encodes a mitochondrial targeting signal, we selected two alternative start sites out of all the predicted non-canonical ones (*Monteuuis et al., 2019*). The ATC codon at position –39 is the most upstream possible start site and the ATG codon is in the middle of the upstream region (*Figure 4G*, left). When we inserted the strong *TEF2pr* one codon before each of the start sites, we observed increased mitochondrial fluorescence of the C-terminally tagged Gpp1-3×GFP₁₁ (*Figure 4G*, right). Interestingly, both Gpp1 and Gpp2 have potential alternative start codons giving rise to an N-terminal extension with high targeting signal prediction, but only Gpp1 has a mitochondrial echoform that we could visualize. This might be due to differential activity of alternative start codons, different strength of targeting signals, or lower expression levels, or more efficient degradation of Gpp2.

To understand the functional impact of having both cytosolic and mitochondrial-matrix isoforms of Gpp1, we performed metabolomic analysis of strains expressing either only the mitochondrial isoform (*TEF2pr*- MTS$_{Su9}$ at canonical ATG that redirects Gpp1 to mitochondria) or only the cytosolic isoform (*TEFpr* at canonical ATG), both (*TEFpr* at –39 alternative start) or, neither (*GPP1* deletion). We quantified a total of 353 soluble metabolites (*Supplementary file 2*) and focused on those involved in glycerol production via the Gpp1/2 pathway (*Figure 4H*). The deletion of Gpp1 did not have any significant

effect on the abundances of dihydroxyacetone-phosphate (DHAP), glyceraldehyde-phosphate (GAP), glycerol-phosphate (3P-Glycerol), and glycerol itself (*Figure 4I*) indicating that Gpp2 and the other pathways can compensate completely in the measured conditions (*Pahlman et al., 2001*). However, native promoter substitution with the strong, overexpressing, *TEFpr* had a strong effect that was independent of driving a cytosolic only or a mitochondrial only isoform: all the glycerol precursors DHAP, GAP, and 3P-glycerol were depleted, and glycerol level was elevated (*Figure 4I*). This result is consistent with the major role of Gpp1 in producing glycerol and suggests that this enzyme performs this function independently of its distribution between cytosol and mitochondria. What the advantage is of having also a mitochondrial isoform is an intriguing question and our work provides the tools to answer it.

## Visualization of protein sorting at the IM

Studying membrane protein biogenesis requires an accurate way to determine topology in vivo. The mitochondrial IM is one of the most protein-rich membranes in the cell supporting a wide variety of TMD topologies with complex biogenesis pathways. We aimed to find out if our BiG Mito-Split collection can accurately visualize the steady-state localization of membrane protein C-termini protruding into the matrix or trap protein transport intermediates.

The most reliable information on protein topology comes from structural studies. To benchmark our data, we selected the structure of the supercomplex of Respiratory Complexes III and IV (*Berndtsson et al., 2020*). In total, the complexes contain six nuclear-encoded subunits whose C-termini face the matrix and all of them are visualized in the BiG Mito-Split collection (*Figure 5A*, shaded green with C-terminus residue number highlighted in black). Interestingly, out of 12 nuclear-encoded subunits with IMS-facing C-termini, we additionally visualize five in disagreement with their expected topology (*Figure 5A*, shaded green with C-terminus residue number highlighted in red). One of them is Rip1 that is well known to be completely imported into the matrix, folded, and only then re-inserted into the IM with the help of the dedicated insertase, Bcs1 (*Wagener et al., 2011*). This suggested that our assay may not visualize only steady-state protein topology but also the biogenesis route of Rip1 as it is exposed to the matrix during the import and assembly process. It was shown that a C-terminal tag on Rip1 can prevent its interaction with the chaperone Mzm1 and promote aggregation in the matrix (*Cui et al., 2012*). It is also possible that our assay visualizes this trapped biogenesis intermediate.

This explanation is not clearly transferable to the other four proteins Cox26, Qcr8, Qcr9, and Qcr10 that are short and have a single TMD. These proteins were suggested to use a stop-transfer mechanism and, therefore, are not expected to be imported into the matrix completely (*Park et al., 2013*). It remains to be determined if matrix translocation can be a part of their normal biogenesis process, followed by the insertion via the conservative pathway. An alternative is that we observe a failure of the stop-transfer mechanism followed by a dead-end or degradation in the matrix. More generally, to date, what other proteins use Bcs1 to insert has remained a mystery (*Wagener and Neupert, 2012*) and it is highly unlikely that this conserved machine evolved only to translocate Rip1. Therefore, our assay may highlight potential new substrates.

We next expanded our analysis and selected all the membrane proteins in our dataset and analyzed their known properties from either the literature (*Supplementary file 3*) or predictions (*Weill et al., 2019*). We compared our observations with the expected steady-state topology (*Figure 5B*, *Figure 5—figure supplement 1*). There were 23 well-studied inner membrane proteins (mono- to hexa-TMDs) in our dataset, 16 of which were expected to have their C-termini facing the matrix and detected in agreement with their steady-state topology (*Figure 5B*, green labels). The other seven were expected to have IMS-facing C-terminus (*Figure 5B*, red labels). Out of those, there were six mono-TMD proteins and one tri-TMD protein Sdh4. The latter was reported to employ a mix of conservative and stop-transfer mechanisms for its biogenesis: the first two TMDs are imported and inserted via Oxa1 while the last TMD is sorted via stop-transfer (*Park et al., 2013*). Two proteins that we uncovered, Spg1 and Nat2, are poorly studied and are possibly IM components (*Figure 5C*, gray labels). Based on the proteomics data (*Morgenstern et al., 2017*) and TMD predictions, we suggest that Spg1 is a mono-TMD protein with its C-terminus facing the IMS and is detected in our experiment like the other short proteins with similar topology. Nat2 most likely is a bi-TMD protein, and we detect its steady-state topology (*Figure 5C*, *Figure 5—figure supplement 1B*).

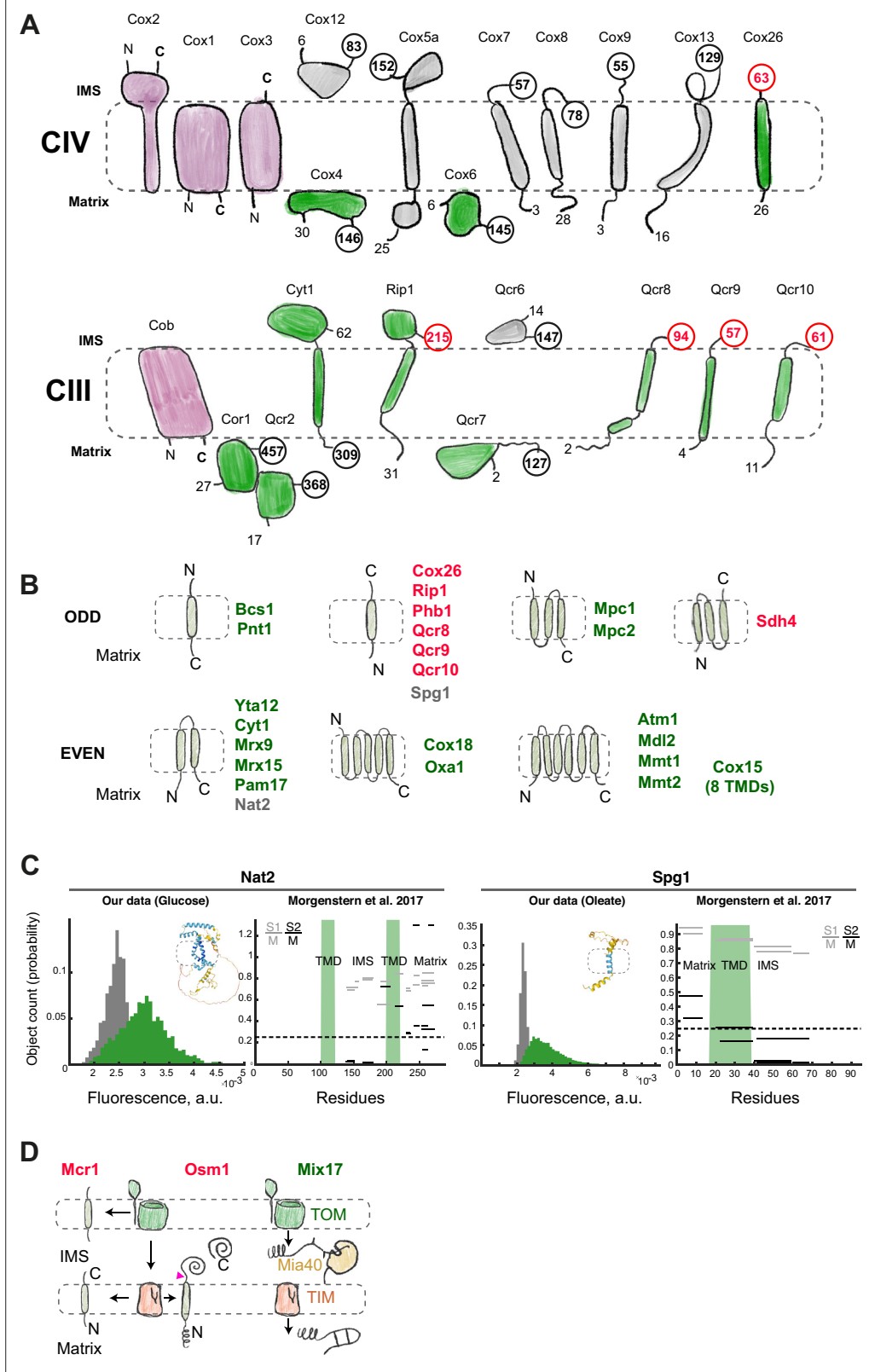

**Figure 5.** Protein topology visualized by BiG Mito-Split. (**A**) Schematic drawing of the topology and approximate sizes of Complex IV and Complex III subunits as seen in a supercomplex structure (PDB:6ymx), mtDNA-encoded subunits are shown in purple, for all other subunits the first and the last amino acid number and position occurring in the structure are shown, C-terminal label is highlighted in bold and enclosed in a circle, proteins that we

*Figure 5 continued on next page*

*Figure 5 continued*

observe in the dataset are colored green, C-terminal label is colored red if the observation does not agree with the known topology. (**B**) All membrane proteins found in our dataset sorted by the number of Transmembrane domains (TMDs) and topology. Those where topology does not agree with our data are marked in red. Those with agreeing topology are marked in green. The two poorly studied proteins are marked in gray. (**C**) Mitochondrial GFP fluorescence intensity for Nat2-3×GFP$_{11}$ and Spg1-3×GFP$_{11}$ shown besides the proteomic data where peptide position and fractionation ratios S1/M and S2/M are shown in gray and black (high S1/M and S2/M mean the peptide is in the matrix; high S1/M and low S2/M mean that the peptide is in the intermembrane space, IMS), TMD prediction is shown in green, AlphaFold2-predicted structures with TMD regions highlighted by dashed lines are shown. (**D**) Other proteins found in the matrix: Mcr1 is a transmembrane protein alternatively sorted into the outer membrane (OM) and the inner membrane (IM), Osm1 is a soluble IMS protein sorted via a stop-transfer mechanism, Mix17 is a possible substrate of Mia40 that is subsequently imported into the matrix.

The online version of this article includes the following figure supplement(s) for figure 5:

**Figure supplement 1.** Comparison of observed inner membrane protein topology with other works and predictions.

---

Interestingly, comparison of our dataset with the expected sub-mitochondrial protein distribution revealed three proteins that were not expected to enter the matrix at all (*Figure 5D*). Mcr1 has a TMD and was described to be alternatively sorted to the outer or the inner membrane (*Hahne et al., 1994*). Osm1 is imported into the IMS and follows the bipartite presequence route (*Neal et al., 2017*). Both proteins require a lateral transfer from the TIM23 complex as a biogenesis step suggesting that we observe occasional mis-sorting of these proteins to the matrix. The third protein is Mix17, CX$_9$C-motif-containing soluble protein annotated to reside in the IMS (*Longen et al., 2009*). Its matrix localization can have physiological significance because it is a homolog of human CHCHD10 that was recently described along with several other similar proteins to contain both a CX$_9$C-motif and a matrix targeting signal (*Peker et al., 2023*). Such proteins are first imported into the IMS, oxidized by Mia40 and then translocated into the matrix, where their oxidative state can regulate macromolecular complex assembly. The only other yeast protein known to follow a similar pathway is Mrp10 (*Longen et al., 2014*). We conclude that the BiG Mito-Split collection is useful to follow protein biogenesis and sorting at the mitochondrial IM as well as unconventional import pathways where a protein can be found in different sub-mitochondrial locations. We suggest that compared to the possible import of cytosolic proteins into the matrix that is prevented by a strict selectivity filter, the selectivity filter at the IM is relatively relaxed.

## Discussion

In this work, we developed a whole-proteome live-cell assay to specifically visualize proteins imported into the mitochondrial matrix or IM. We imaged ~50 proteins not observed as mitochondrial by microscopy before. Some of them were poorly studied, eclipsed proteins that were localized in the cytosol or nucleus by previous studies (*Supplementary file 1*). Finding these proteins using the BiG Mito-Split assay was expected. However, we managed to image some proteins that were never imaged before using traditional full-length FP fusions. We argue that our collection of 3×GFP$_{11}$-tagged proteins can offer an advantage over the previous ones by using a smaller unfolded tag that may interfere less with protein targeting. It might be particularly useful for mitochondrial proteins for which it is long known that a tightly folded domain can inhibit their translocation (*Wienhues et al., 1991*). Thus, full-length FPs that are optimized for folding and maturation speed can be particularly problematic tags for mitochondrial proteins (*Chudakov et al., 2010*).

How sensitive is the BiG Mito-Split assay? Since the annotation of sub-mitochondrial protein localization and topology is incomplete, it is hard to make a good estimate. Here, we find 70% of proteins that were reported to be soluble in the matrix by fractionation and proteomics (*Figure 3—figure supplement 1A*; *Vögtle et al., 2017*). While low abundance proteins are probably more likely to be missed, we still find a lot of them (*Figure 3B*) meaning that there is no strict sensitivity threshold for the BiG Mito-Split assay. We suggest that the major factor contributing to the assay sensitivity is the interaction probability of the two split-GFP fragments. The GFP$_{1-10}$/GFP$_{11}$ fragments do not have high affinity to each other and are commonly used to study protein-protein interactions (*Romei and*

*Boxer, 2019*). Indeed, in our dataset, some matrix proteins related to mitochondrial translation have higher fluorescence signal than expected from their abundance (*Figure 3—figure supplement 1B*), probably owing to a higher probability of interacting with the newly synthesized GFP$_{1-10}$. Using split-reporters with higher affinity between the fragments can make the BiG Mito-Split more sensitive. The other possible reasons for not observing particular proteins with our assay are inherent to the incompleteness of yeast strain collections and interference of tags with protein expression, stability, or targeting. We did not find that protein complex components or essential proteins are more likely to be false negatives. However, some essential proteins were absent from the collection to start with (*Figure 3—figure supplement 2A*). Thus, a small tag allows visualization of even complex proteins. In this version of the assay, the correct tagging and expression of each protein needs to be confirmed by PCR and western blots. Developing collections where protein expression can be independently controlled with fluorescence in another channel can help to get better estimates of sensitivity and to interpret the absence of signal.

Based on the comparison of our dataset with the reported protein abundances, we concluded that the detection of low abundance protein fractions is possible. We wondered if, at any of the tested conditions, some cytosolic proteins that we found could be imported by chance or mistake. We suggest that cytosolic r-proteins are among those. They were only detected in a few conditions at low levels. Most of them were found on logarithmic growth phase on glucose and almost none in stationary phase (*Supplementary file 1*) despite many other proteins that have higher signals (*Figure 2A*). We suggest that this is due to high synthesis rate of new ribosomal proteins during quick growth on glucose and due to lower rate of their synthesis and upregulation of mitochondrial quality control in stationary phase. There are hundreds of proteins that similarly to r-proteins contain regions that are predicted as mitochondrial targeting signals but only few are imported (*Figure 3C*). Moreover, proteotoxic stress was suggested to trigger unspecific uptake of aggregation-prone cytosolic proteins for their subsequent degradation (*Ruan et al., 2017*). We imaged the BiG Mito-Split collection in several conditions, including DTT-induced ER stress but did not detect any large group of proteins that becomes conditionally imported (*Figure 2A*, *Supplementary file 1*). We conclude that under the conditions we investigated, mitochondria maintain a strict selectivity filter at the OM. It was also recently shown that the r-protein uS5 (encoded by RPS2 in yeast) has a latent MTS that is masked by a special mitochondrial avoidance segment (MAS) preceding it (*Oborská-Oplová et al., 2025*). The removal of the MAS leads to import of uS5 into mitochondria killing the cells. The case of uS5 is an example of occasional similarity between an r-protein and an MTS caused by similar requirements of positive charges for rRNA binding and mitochondrial import. It remains unclear if other r-proteins have a MAS and if there are other mechanisms that protect mitochondria from translocation of cytosolic proteins. Also, the specific function for some r-proteins in mitochondria cannot be completely ruled out and requires more detailed investigation. We suggest that the whole-genomic assay that we developed will be useful to study the mechanisms of the mitochondrial import specificity and to clarify what are the conditions when the selectivity is compromised.

Using the BiG Mito-Split assay, we were also able to visualize IM proteins with their C-termini exposed to the matrix. Interestingly, we revealed several proteins that were suggested to have a different topology. This includes several single-TMD proteins and one 3-TMD protein (Sdh4) that were reported to use stop-transfer mechanism and to be arrested in the TIM23 complex and laterally released into the membrane. We also found one protein that uses this mechanism for import into the IMS (Osm1). One possibility is that we observe productive import intermediates that can be later reinserted into the IM by the Oxa1 translocase. Only a few nuclear-encoded substrates were suggested to rely on the Oxa1 translocase for reinsertion into the IM via the conservative pathway (*Bohnert et al., 2010*; *Park et al., 2013*; *Stiller et al., 2016*). Now it is becoming clear that the substrate range of Oxa1 might be much broader (*Stiller et al., 2016*). The mechanisms that Oxa1 uses for protein insertion are similar to the ER Membrane Complex (EMC), its distant homolog (*Kizmaz and Herrmann, 2023*). In vivo fluorescence assays were among the most important tools that helped to gain considerable insight into how EMC selects its substrates and ensures correct TMD orientation within the membrane (*Wu et al., 2024*; *Pleiner et al., 2023*; *Fenech et al., 2023*). We suggest that BiG Mito-Split can be an excellent tool to address these mechanisms for Oxa1. It is possible that the proteins with C-termini that are translocated into the IMS from the matrix side can be trapped by the interaction with GFP$_{1-10}$. In that case, our assay can be a useful tool to study these pre-translocation

intermediates. The observation of Osm1 translocation into the matrix raises another interesting possibility that TIM23 complex is not very selective in discriminating between the substrates that need to be arrested and laterally sorted, and those that need to pass through. In this case, our assay shows considerable flexibility of the protein sorting at the IM as opposed to the strict selectivity filter at the OM. The BiG Mito-Split collection will be helpful to study whether mis-sorting of IM proteins to the matrix is deleterious to the cells and which quality control mechanisms are responsible for managing it.

To conclude, using a new generation of yeast collections (*Meurer et al., 2018*; *Yofe et al., 2016*), we brought a fluorescence complementation assay (*Bader et al., 2020*) to a whole-genomic level. In this work, we applied it to systematic visualization of mitochondrial matrix proteome by employing a matrix-localized GFP$_{1-10}$. We used this tool to uncover new mitochondrial-localized proteins and create a dataset and a resource for the mitochondrial protein field to study protein import and topogenesis in vivo. More broadly, our new SWAT GFP$_{11}$ library can be utilized for many other purposes. Beyond the BiG Mito-Split assay, by targeting the GFP$_{1-10}$ to other cellular destinations, the collection can be used to visualize targeting processes in additional organelles. In addition, it can be utilized to perform systematic protein-protein interaction studies. As such, our library provides a powerful tool for systematic probing of cellular architecture.

# Materials and methods

**Key resources table**

| Reagent type (species) or resource | Designation | Source or reference | Identifiers | Additional information |
|---|---|---|---|---|
| Genetic reagent (*S. cerevisiae*) | SWAT donor | *Yofe et al., 2016* | YMS2085 | |
| Genetic reagent (*S. cerevisiae*) | MTS-mCherry library donor | This study; based on YMS2085 | YMS6308 | Used to make haploid library; available on request from the corresponding authors |
| Genetic reagent (*S. cerevisiae*) | BIG-SPLIT-Matrix-GFPß1–10 | *Bader et al., 2020* | RKY176 | |
| Genetic reagent (*S. cerevisiae*) | BY4741 | *Brachmann et al., 1998* | BY4741 | |
| Genetic reagent (*S. cerevisiae*) | BIG-SPLIT-Matrix-GFPß1–10 BY4741 | This study | RKY250(YMS5839) | Available on request from the corresponding authors |
| Genetic reagent (*S. cerevisiae*) | C'-SWAT acceptor library | *Meurer et al., 2018* | Schuldiner lab | Used to make haploid library |
| Recombinant DNA reagent | TEFpr-MTS-mCherry-MET | This work | pMS1318 | For PCR-based insertion into genomic locus and expression from there; available on request from the corresponding authors |
| Recombinant DNA reagent | pFA6 NAT | *Longtine et al., 1998* | pMS1214 | For PCR-based mutagenesis |
| Recombinant DNA reagent | C-SWAT type I | *Meurer et al., 2018* | pMS808 | Empty vector to create C-SWAT donor plasmid to swap acceptor cassette with a donor tag restoring a native gene terminator without selection marker |
| Recombinant DNA reagent | 3xGFP11 C-SWAT donor type I | This work | pMS1263 | Based on pMS808, 3xGFP11 C-SWAT donor plasmid to swap acceptor cassette with a donor tag restoring a native gene terminator without selection marker; available on request from the corresponding authors |
| Recombinant DNA reagent | 3xGFP11 C-SWAT donor type III | This work | pMS1264 | 3xGFP11 C-SWAT donor plasmid to swap acceptor cassette with a donor tag, followed by ADHter and KanMX, can be used for PCR-based mutagenesis using pYM linkers; available on request from the corresponding authors |
| Antibody | Anti-GFP (Mouse Monoclonal IgG$_1$κ clones 7.1 and 13.1) | Roche | Cat# 11814460001 | WB (1:5000) recognizes GFP$_{11}$ |
| Chemical compound, drug | MitoTracker Red CMXRos | ThermoFisher | Cat# M7512 | Mitochondria staining |

## Yeast strains and plasmids

The yeast strains used in this study are listed in *Supplementary file 4*, Plasmids are listed in *Supplementary file 5*, and primers used for yeast strain construction are listed in *Supplementary file 6*.

To make the BiG Mito-Split-GFP assay (*Bader et al., 2020*) compatible with the library genetic background, a new strain RKY250 encoding $GFP_{1-10}$ in the mtDNA was constructed. For this, the RKY176 ($\rho^+$ atp6::$GFP_{1-10}$ 5`$UTR_{COX2}$ ATP6 3`$UTR_{COX2}$) was crossed to BY4741 [$\rho°$]. The diploid cells were selected on synthetic complete medium supplemented with leucine and uracil and sporulated in liquid 1% potassium acetate for one week. Tetrads were dissected and analyzed for their genotype. The presence of the mtDNA $\rho^+$ atp6::$GFP_{1-10}$ 5`$UTR_{COX2}$ ATP6 3`$UTR_{COX2}$ was verified in each spore by checking the GFP fluorescence signal with $ATP4-GFP_{11}$ expressed from the plasmid pAG416pGPD-ATP4-$GFP_{11}$. This plasmid was obtained by transferring $ATP4-GFP_{11}$ from the original pAG414 *pGPD-ATP4-$GFP_{11}$* (*Bader et al., 2020*) to the pAG416 *pGPD-ccdB* with the Gateway assembly cloning method (*Alberti et al., 2007*).

The SWAT donor strain used to make the new $C'–3×GFP_{11}$ SWAT library was constructed on the basis of the published *GAL1pr-SceI* strain (*Yofe et al., 2016*). To visualize mitochondria, we inserted $MTS_{Su9}$-mCherry from the plasmid TEFpr-MTS-mCherry-MET into the *HO* locus. The plasmid was made by replacing mTagBFP2 in plasmid #44899 from AddGene (*Lee et al., 2013*) with TEF2pr-Su9MTS-mCherry and the KanMX with MetR from pSD-N21 (*Weill et al., 2018*). A NAT resistance cassette was inserted into the *URA3* locus to aid diploid cell selection on rich respiratory media (see below).

The plasmid bearing a $3×GFP_{11}$ cassette for the SWAT library construction ($3×GFP_{11}$ C-SWAT type I, no additional terminator and selection marker after the tag) was created by amplifying $3×GFP_{11}$ from the GPDpr-PGK1-$3×GFP_{11}$ plasmid (*Bader et al., 2020*) with added BamHI and BcuI(SpeI) restriction sites, and performing digest and ligation into the C-SWAT type I donor plasmid (*Meurer et al., 2018*). The plasmid $3×GFP_{11}$-ADHter-KanMX used for genomic tagging with $3×GFP_{11}$ was created by inserting the same amplified $3×GFP_{11}$ fragment into the C-SWAT type III plasmid (contains *ADH1* terminator and KanMX gene following the inserted tag) using the same restriction enzymes.

The strains to investigate the targeting of Gpp1 and Gpp2 were created by genomically tagging *GPP1* and *GPP2* with $3×GFP_{11}$-*ADH*ter-KanMX using this type III plasmid. Since this plasmid contains CEN/ARS, the standard PCR-mediated transformation protocol was modified with an additional DpnI digest (37°C, 1.5 hr, directly in the PCR buffer) of the amplified cassette to destroy the original plasmid and only allow transformation by genomic integration, and not by plasmid propagation. Transformants were verified by PCR and microscopy. To modify the promoter region of $GPP1–3×GFP_{11}$ and $GPP2–3×GFP_{11}$ strains, a regular promoter swap was performed using pYM N-19 *NAT-TEF2pr, pYM N-20 NAT-TEF2pr-3HA*, or *N-SWAT NAT-MTS(Su9)–3HA* plasmid. Both the change of promoter and the elimination of the gene with native promoter was confirmed by PCR screening to avoid gene duplication events.

For visualization of Arc1 and Cha1, $3×GFP_{11}$ epitope-tagging, transformation, growth of the transformed RKY176 strain, and verification of both the tagged protein and $GFP_{1-10}$ expression were performed as described in *Hemmerle et al., 2022*. The strains where Ilv6, Fum1, and Arc1 are tagged with mNeonGreen were picked from the C-SWAT library prepared exactly like described (*Meurer et al., 2018*).

All strain modifications were performed using the standard PCR-mediated methods of genomic editing and LiAc-based yeast transformation (*Longtine et al., 1998*; *Janke et al., 2004*).

## Creation of C-SWAT 3xGFP$_{11}$ haploid collection and diploid BiG-Split collection

To create the new C-SWAT collection where every gene is tagged with $3×GFP_{11}$, we transformed the $MTS_{Su9}$-mCherry donor strain with the $3×GFP_{11}$ plasmid selecting the transformants on G418-containing media (1% yeast extract, 2% peptone, 3% glucose, 500 µg/l geneticin, or G418 (Formedium)). Library creation was performed using this strain essentially as described before (*Meurer et al., 2018*; *Weill et al., 2018*) using automated mating and selection procedures (*Tong and Boone, 2007*; *Cohen and Schuldiner, 2011*) and robotic arrayer Rotor HDA (Singer Instruments). In brief, the donor strain was mated with the C-SWAT acceptor library (*Meurer et al., 2018*). The resulting diploids were selected and sporulated. After sporulation, haploids of mating type alpha were selected that were carrying the $3×GFP_{11}$ SWAT plasmid. The tag swapping was induced by plating the libraries

on galactose-containing medium. The strains where the acceptor SWAT cassette was successfully swapped with 3×GFP$_{11}$ were negatively selected on 5-FOA-containing media (Formedium) to kill the cells still containing the *URA3* marker in the acceptor cassette. Recombination efficiency was verified by PCR in a random set of strains and was found to be more than 90%. The resulting haploid library had every gene genomically tagged with 3×GFP$_{11}$ followed by the native terminator without any additional selection markers (seamless tagging).

To obtain the diploid library that had both genomic 3×GFP$_{11}$ tags and mtDNA-encoded GFP$_{1-10}$, the haploid library was first depleted of its own mtDNA by plating three times on 3% agar plates synthetic dextrose media 0.67% yeast nitrogen base with ammonium sulfate, 2% glucose, complete amino acid mix optimized for BY4741 background (*Hanscho et al., 2012*) supplemented 25 µg/ml ethidium bromide. To check that the depletion was successful, the library was plated on rich respiratory media (YPGly) to confirm there is no growth in these conditions (1% yeast extract, 2% peptone, 3% glycerol). After mtDNA depletion, the haploids were immediately mated with the RKY250 BiG-Split strain. The diploids were selected and maintained on YPGly supplied with 200 µg/l nourseothricin (NAT, Jena Bioscience). Several diploids isogenic to the library and containing mtDNA-encoded GFP$_{1-10}$ but no GFP$_{11}$ were added to each plate as controls. The resulting diploid library was used for fluorescent microscopy screening.

## Whole-genomic (primary) screening

The whole diploid library was imaged in six different conditions. All the sample preparation was performed with a robotic liquid handler Freedom EVO (Tecan) equipped with an incubator and a centrifuge. Before each experiment, the cells were transferred from agar plates to liquid synthetic dextrose (SD) medium without methionine supplied with NAT and grown overnight at 30°C. The overnight culture was then subjected to different conditions. For imaging of logarithmically growing cells, the overnight culture was diluted 1:20 in fresh SD media supplied with complete amino acid mix and grown for 4 hr at 30°C. The resulting culture was applied onto glass-bottom microscopy plates, cells were let to settle for 25 min and then washed one time: the media with unattached cells was aspirated, and fresh media was added. For imaging in stationary phase, the overnight culture was directly applied onto microscopy plates (Matrical Biosciences) and washed three times with synthetic ethanol media (0.67% yeast nitrogen base, 2% ethanol). For imaging in low phosphate conditions, the overnight culture was centrifuged at 3000 g for 3 min, supernatant was aspirated and the cells resuspended in distilled water, centrifugation and resuspension was repeated two more times. The resulting cell suspension was diluted 1:100 in synthetic dextrose medium without phosphate (SD-P$_i$, 0.67% yeast nitrogen base without inorganic phosphate, 2% glucose) and incubated for 16 hr at 30°C. Then the cells were transferred to microscopy plates as described above and washed one time with SD-P$_i$. For imaging in DTT-containing media, the cells were prepared the same way as for imaging in SD media, except that the media contained 1 mM DTT, and the growth before imaging was limited to 3 hr. For imaging in nitrogen starvation media (SD-N, 0.67% yeast nitrogen base without ammonium sulfate, 2% glucose), the overnight culture was first diluted 1:20 in SD media and grown for 4 hr at 30°C. Then the cells were pelleted by centrifugation, the media removed, and substituted with SD-N. Centrifugation and media exchange was repeated one more time and after this, the cells were incubated for 16 hr at 30°C, applied onto microscopy plates, washed with SD-N and imaged. For imaging in oleate media, the cells were grown overnight as above and then diluted 1:20 in synthetic oleate media SOleate, 0.67% yeast nitrogen base 0.2% oleic acid, 0.1% Tween 80. All the imaging was performed using an Olympus SpinSR system equipped with Hamamatsu flash Orca 4.0 camera and a CSUW1-T2SSR SD Yokogawa spinning disk unit with a 50 µm pinhole disk, and a 60x air lens with NA 0.9. Images were recorded with 488 nm laser illumination for GFP channel, 561 nm laser illumination for mCherry. The microscope was operated by ScanR Acquisition software (version 3.2). Micrographs were visualized using Fiji (*Schindelin et al., 2012*). The image of each strain was visually compared with a micrograph of the control strain from the same plate. The strains with increased signal in at least one condition, or the strains where the signal increase was ambiguous were selected for a secondary screen to measure fluorescence quantitatively (total 543 strains).

## Fluorescence quantitation (secondary screen)

The 543 strains selected in the primary screen were arrayed on two 384-format plates. To these strains, we added 123 non-fluorescent control strains positioned evenly across each plate and 102 additional strains that were not selected in the primary screens but were reported to reside in the matrix or the IM, based on the two most recent and reliable proteomic studies (*Morgenstern et al., 2017*; *Vögtle et al., 2017*). The strains were prepared for imaging exactly as described in the previous section. The imaging was performed with the same automated SpinSR system but using a 100x air lens with NA 1.3 equipped with an oil pump. The data was analyzed using CellProfiler v. 4.2.1+ (*Stirling et al., 2021*). The image analysis pipeline was constructed to perform background subtraction in both GFP and mCherry channels, enhance small objects in mCherry channel, segment mitochondrial objects using the mCherry channel and measure the median GFP signal intensity within each mitochondrial object. The final dataset contained fluorescence signal measured for each mitochondrion in each strain and condition. This data was further analyzed in MATLAB (Mathworks). Median intensity of all the segmented mitochondria was used as a raw signal measurement for each strain. Since the variability of the signal was high even among the control strains without $GFP_{11}$, we designed a score to represent how much median mitochondrial GFP signal in $GFP_{11}$-containing strains deviates from non-fluorescent controls. For this, we subtracted the average control fluorescence from all the values and divided them by standard deviation of all controls from this average. Less than 1% of control strain measurements were above 3 standard deviations (*Figure 2—figure supplement 1A*). We designated every $GFP_{11}$-containing strain that has a signal exceeding control average by 5 standard deviations as a confident hit, and by more than 3 and less than 5 standard deviations as an ambiguous hit (*Supplementary file 1*). Most ambiguous hits were still visually well distinguishable from controls (*Figure 2—figure supplement 1B*). Most of the strains were confident hits in all the conditions (*Figure 2—figure supplement 1C*). The strains that were hits only in a few conditions were most likely missed in the others due to low signal intensity and not due to strong differential regulation of protein expression or import (*Figure 2B*). Some conditions had less hits, most probably due to high media autofluorescence (*Figure 2—figure supplement 1D*). Several strains were imaged one more time in all conditions to confirm the observation (*Supplementary file 1*). Thus, we compiled a unified list of proteins found in this study (*Supplementary file 1*, marked as 'OBSERVED'). The complete image dataset used for quantification is available on the BioImage Archive (https://www.ebi.ac.uk/bioimage-archive/) accession number S-BIAD2409.

## Fluorescence microscopy

All microscopy of individual strains with genomically integrated tags was performed as described above for fluorescence quantitation, except each strain was grown individually in appropriate media. To visualize the potential mitochondrial import of Cha1 and Arc1, we used the Zeiss LSM 800 with Airyscan of the imaging platform of the Research Center in Biomedicine of Strasbourg (PIC-STRA, CRBS). A 63× apochromatic plane objective (1.40 digital aperture) with oil immersion, LED illumination (Colibri, Zeiss) and GFP and DsRED filter cubes were used for direct observation using eyepieces. In confocal mode, 488 nm and 640 nm lasers replaced the LEDs and high sensitivity photomultipliers (GaAsP, Phospho Arsenide de Gallium) for detection of the emitted fluorescence. The images are acquired using the Zeiss Zen Blue software, with a laser residence time per pixel (pixel dwell time) of 11.8 µs. For visualization, the image contrast was linearly adjusted in Fiji and the images were converted from 16-bit to 8-bit (*Schindelin et al., 2012*).

## Protein extraction and western blot

Protein extraction, SDS-PAGE, and western blotting were performed as described before with the same antibodies against $GFP_{11}$ and $GFP_{1-10}$ (*Bader et al., 2020*). Adh1 was probed with rabbit polyclonal primary antibody which was a gift from Claudio De Virgilio's lab (Université de Fribourg, Switzerland) used with a dilution of 1:50,000 (Calbiochem ref 126745). The secondary antibody was Goat-anti-rabbit-HRP conjugate used with a dilution of 1:5000.

## Sample Preparation for Metabolomics

Starter cultures were generated by inoculating 3 mL of YPD (with appropriate selections) with each strain in triplicate. Cultures were incubated overnight at 30°C with shaking. Subsequently, 50 µL of the

overnight culture was used to inoculate 5 mL of fresh YPD, followed by another overnight incubation at 30°C on a shaker. Cultures were then scaled up to 50 mL and grown in YP medium supplemented with 2% ethanol until reaching an $OD_{600}$ of 0.5, followed by a 4 hr incubation. Cells were harvested by centrifugation at 3000 rpm for 3 min, washed with 1 mL of double-distilled water (DDW), centrifuged again, and the supernatant was removed. The resulting cell pellets were snap-frozen in liquid nitrogen.

## Metabolite Extraction

Polar metabolites were extracted and analyzed based on the protocols from *Malitsky et al., 2016* and *Zeng et al., 2015*, with modifications. Cell pellets were extracted using 1 mL of pre-chilled (−20°C) methanol:MTBE (1:3, v/v). Samples were vortexed and then sonicated for 30 min in an ice-cold sonication bath, with brief vortexing every 10 min. A 0.5 mL solution of DDW:methanol (3:1, v/v) containing an internal standard mix of ^13C- and ^15N-labeled amino acids (Sigma, 767964) was added. After centrifugation, the upper organic phase was collected into a 2 mL Eppendorf tube. The polar phase was re-extracted with 0.5 mL MTBE, and the resulting organic phase was pooled with the first. The remaining polar phase was transferred to a fresh tube, dried under nitrogen for 1 hr to remove organic solvents, and then lyophilized. Prior to LC-MS analysis, samples were reconstituted in 150 μL of DDW:methanol (1:1), centrifuged twice at 13,000 rpm to remove precipitates, and transferred to HPLC vials.

## LC-MS Analysis of Polar Metabolites

Metabolite profiling was carried out following the method described by *Zeng et al., 2015*, with slight modifications. An Acquity I-Class UPLC system coupled with a Q Exactive Plus Orbitrap mass spectrometer (Thermo Fisher Scientific) was used in negative ionization mode. Separation was performed using a SeQuant Zic-pHilic column (150 mm × 2.1 mm) with a guard column (20 mm × 2.1 mm) from Merck. Mobile phase B consisted of acetonitrile, and mobile phase A comprised 20 mM ammonium carbonate with 0.1% ammonia hydroxide in DDW:acetonitrile (80:20, v/v). The flow rate was maintained at 200 μL/min with the following gradient: 0–2 min, 75% B; 14 min, 25% B; 18 min, 25% B; 19 min, 75% B; followed by 4 min re-equilibration.

## Statistical Analysis of Metabolomics Data

To compare metabolite levels before and after protein depletion, two-tailed t-tests assuming equal variances (homoscedastic) were performed. The resulting data tables include normalized values, metabolite names, relative abundances, identification levels, and chemical formulas. Additionally, visualizations include average values with standard error bars, results of basic statistical analysis (t-test), Principal Component Analysis (PCA), and identification of potential co-eluting compounds sharing similar masses, fragmentation patterns, and retention times.

## Additional data analysis

Targeting signal predictions for each protein and its alternatively translated form were taken from *Monteuuis et al., 2019*. We always chose the highest score from the three different programs that were used in this work. Internal targeting sequences were predicted using the i-MLP web server (http://csb-imlp.bio.rptu.de/) (*Boos et al., 2018*). Unified data on protein abundance for *Figure 2—figure supplement 1B* was taken from *Ho et al., 2018* or *Morgenstern et al., 2017*. All plots were produced using MATLAB (Mathworks).

## Acknowledgements

We would like to thank Tamara Flohr for critical reading of the manuscript and Dr. S Friant (CNRS, UMR 7156) for her help in confocal microscopy imaging. We are grateful to Claudio De Virgilio (Université de Fribourg, Switzerland) for kindly sharing antibodies. We thank Christian Koch and Aleksandr Kirillov with the help with cloning and Sabine Knaus for technical assistance and Felix Boos for sharing the data on alternative translation start in Gpp1. Work in the Schuldiner lab is supported by the German Research Foundation (DFG) collaborative grant # 1028/11–1 and an ERC CoG OnTarget (864068). YB was supported by the EMBO Long-term postdoctoral fellowship (ALTF 480–2019). MS is an incumbent of the Dr Gilbert Omenn and Martha Darling Professorial Chair in Molecular Genetics. The robotic system of the Schuldiner lab was purchased through the kind support of the Blythe Brenden-Mann Foundation. Work in the Kucharczyk lab was supported by the NSC grant nr 2018/31/B/NZ3/01117.

The work in the Becker lab was supported by the Integrative Molecular and Cellular Biology (IMCBio), as part of the Interdisciplinary Thematic Institutes (ITI) 2021-to-2028 program of the University of Strasbourg, CNRS, and INSERM, supported by IdEx Unistra (ANR-10-IDEX-0002) and EUR IMCBio (ANR-17-EURE-0023) under the framework of the French Investments for the Future Program (to SZ, HDB, BS); by the University of Strasbourg (to SZ, HDB, BS), by the CNRS (to SZ, HDB, BS).

## Additional information

### Funding

| Funder | Grant reference number | Author |
|---|---|---|
| European Research Council | 864068 | Maya Schuldiner |
| European Molecular Biology Organization | ALTF 480-2019 | Yury S Bykov |
| Centre National de la Recherche Scientifique | Thematic Institutes (ITI) 2021-to-2028 | Hubert D Becker |
| National science center | 2018/31/B/NZ3/01117 | Roza Kucharczyk |
| IdEx Unistra | ANR-10-IDEX-0002 | Hubert D Becker |
| EUR IMCBio | ANR-17-EURE-0023 | Hubert D Becker |
| German Research Foundation | 1028/11–1 | Maya Schuldiner |

The funders had no role in study design, data collection and interpretation, or the decision to submit the work for publication.

### Author contributions

Yury S Bykov, Conceptualization, Data curation, Formal analysis, Investigation, Writing – original draft, Writing – review and editing; Solene Zuttion, Johanna Arnold, Formal analysis, Investigation, Methodology, Writing – review and editing; Dunya Edilbi, Marina Polozova, Formal analysis, Investigation, Writing – review and editing; Sergey Malitsky, Maxim Itkin, Data curation, Formal analysis, Investigation, Methodology, Writing – review and editing; Bruno Senger, Hadar Meyer, Investigation, Methodology, Writing – review and editing; Ofir Klein, Resources, Methodology; Yeynit Asraf, Software, Investigation, Methodology, Writing – review and editing; Hubert D Becker, Conceptualization, Supervision, Funding acquisition, Investigation, Project administration, Writing – review and editing; Roza Kucharczyk, Conceptualization, Funding acquisition, Investigation, Methodology, Writing – original draft, Project administration; Maya Schuldiner, Conceptualization, Supervision, Funding acquisition, Investigation, Writing – original draft, Project administration, Writing – review and editing

### Author ORCIDs

Yury S Bykov (ID) https://orcid.org/0000-0003-2959-4108
Marina Polozova (ID) https://orcid.org/0000-0001-6678-1688
Sergey Malitsky (ID) https://orcid.org/0000-0003-4619-7219
Maxim Itkin (ID) https://orcid.org/0000-0003-1348-2814
Bruno Senger (ID) https://orcid.org/0000-0002-5157-7307
Ofir Klein (ID) https://orcid.org/0000-0002-9635-4481
Hadar Meyer (ID) https://orcid.org/0000-0001-5725-0447
Hubert D Becker (ID) https://orcid.org/0000-0002-4102-7520
Roza Kucharczyk (ID) https://orcid.org/0000-0002-8712-7535
Maya Schuldiner (ID) https://orcid.org/0000-0001-9947-115X

Reviewer #1 (Public review): https://doi.org/10.7554/eLife.98889.3.sa1
Reviewer #2 (Public review): https://doi.org/10.7554/eLife.98889.3.sa2
Reviewer #3 (Public review): https://doi.org/10.7554/eLife.98889.3.sa3
Author response https://doi.org/10.7554/eLife.98889.3.sa4

# Additional files

## Supplementary files

Supplementary file 1. The complete dataset of normalized fluorescence in each condition and comparisons with other data. Full legend inside file.

Supplementary file 2. The quantification of polar metabolite abundances in the strains with mutated Gpp1. Full legend inside file.

Supplementary file 3. Comparison of known inner membrane protein topology with the observations made using BiG Mito-Split collection. Full legend inside file.

Supplementary file 4. Yeast strains used in this work.

Supplementary file 5. Plasmids used in this work.

Supplementary file 6. Primers used for yeast strain construction.

MDAR checklist

## Data availability

The yeast strains and strain collections generated in this work are available from the corresponding authors on request. The source data is included in the manuscript, except for the high-content imaging data used for fluorescence quantification. This dataset is deposited to the BioImage Archive, accession number S-BIAD2409.

The following dataset was generated:

| Author(s) | Year | Dataset title | Dataset URL | Database and Identifier |
|---|---|---|---|---|
| Bykov YS, Zuttion S, Edilbi D, Polozova M, Arnold J, Malitsky S, Itkin M, Senger B, Klein O, Asraf Y, Meyer H, Becker HD, Kucharczyk R, Schuldiner M | 2025 | Identification of mitochondrial matrix-facing proteins by high-throughput imaging of the diploid BiG Mito-Split yeast library | https://www.ebi.ac.uk/biostudies/bioimages/studies/S-BIAD2409 | BioImage Archive, S-BIAD2409 |

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
