## [Editor Report · eLife Assessment]

This study represents a **valuable** addition to the catalog of mitochondrial proteins. With the use of methodology based on the bi-genomic split-GFP technology, the authors generate **convincing** data, including dually localized proteins and topological information, under various growth conditions in yeast. The study represents a key basis for further functional and/or mechanistic studies on mitochondrial protein biogenesis.

---

## [Referee Report · Reviewer #1 (Public review)]

Summary:

The study conducted by the Schuldiner's group advances the understanding of mitochondrial biology through the utilization of their bi-genomic (BiG) split-GFP assay, they had previously developed and reported. This research endeavors to consolidate the catalog of matrix and inner membrane mitochondrial proteins. In their approach, a genetic framework was employed wherein a GFP fragment (GFP1-10) is encoded within the mitochondrial genome. Subsequently, a collection of strains was created, with each strain expressing a distinct protein tagged with the GFP11 fragment. The reconstitution of GFP fluorescence occurs upon the import of the protein under examination into the mitochondria.

Strengths:

Notably, this assay was executed under six distinct conditions, facilitating the visualization of approximately 400 mitochondrial proteins. Remarkably, 50 proteins were conclusively assigned to mitochondria for the first time through this methodology. The strains developed and the extensive dataset generated in this study serve as a valuable resource for the comprehensive study of mitochondrial biology. Specifically, it provides a list of 50 "eclipsed" proteins whose role in mitochondrial remains to be characterized.

The work could include some functional studies of the dually localized Gpp1 protein, as an example.

---

## [Referee Report · Reviewer #2 (Public review)]

The authors addressed the question how mitochondrial proteins that are dually localized or only to a minor fraction localized to mitochondria can be visualized. For this they used an established and previously published method called BiG split-GFP, in which GFP strands 1-10 are encoded in the mitochondrial DNA and fused the GFP11 strand C-terminally to the yeast ORFs using the C-SWAT library. The generated library was imaged under different growth and stress conditions and yielded positive mitochondrial localization for approximately 400 proteins. The strength of this method is the detection of proteins that are dually localized with only a minor fraction within mitochondria, which was so far has hampered due to strong fluorescent signals from other cellular localizations. The weakness of this method is that due to the localization of the GFP1-10 in the mitochondrial matrix, only matrix proteins and IM protein with their C-termini facing the matrix can be detected. In addition, The C-terminal GFP11 might impact on assembly of proteins into multimeric complexes or interfere with biogenesis trapping the tagged protein in an unproductive transport intermediate. Taken these limitations into consideration, the authors provide a new library that can help in identification of eclipsed protein distribution within mitochondria, thus further increasing our knowledge on the complete mitochondrial proteome. The approach of global tagging of the yeast genome is the logical consequence after the successful establishment of the BiG split-GFP for mitochondria. The authors also propose that their approach can be applied to investigate the topology of inner membrane proteins, however, for this the inherent issue remains that even the small GFP11 tag can impact on protein biogenesis and topology. Thus, the approach will not overcome the need to assess protein topology via biochemical approaches detecting endogenous untagged proteins.

Comments on revisions:

The first sentence of the abstract should be changed as the statement that "The majority of the mitochondrial proteins (...) often lack clear targeting signals" is in particular for the here analysed IM and matrix protein not correct: Several N-proteomics analysis have defined N-terminal cleavable targeting signals in great detail.

Also the statement in the title that the assay illuminates protein targeting routes should be reconsidered as experimental evidence for this statement is still scarce.

---

## [Referee Report · Reviewer #3 (Public review)]

Summary:

Here, Bykov et al move the bi-genomic split-GFP system they previously established to the genome-wide level in order to obtain a more comprehensive list of mitochondrial matrix and inner membrane proteins. In this very elegant split-GFP system, the longer GFP fragment, GFP1-10, is encoded in the mitochondrial genome and the shorter one, GFP11, is C-terminally attached to every protein encoded in the genome of yeast *Saccharomyces cerevisiae*. GFP fluorescence can therefore only be reconstituted if the C-terminus of the protein is present in the mitochondrial matrix, either as part of a soluble protein, a peripheral membrane protein or an integral inner membrane protein. The system, combined with high-throughput fluorescence microscopy of yeast cells grown under six different conditions, enabled the authors to visualize ca. 400 mitochondrial proteins, 50 of which were not visualised before and 8 of which were not shown to be mitochondrial before. The system appears to be particularly well suited for analysis of dually localized proteins and could potentially be used to study sorting pathways of mitochondrial inner membrane proteins.

Strengths:

Many fluorescence-based genome-wide screen were previously performed in yeast and were central to revealing the subcellular location of a large fraction of yeast proteome. Nonetheless, these screens also showed that tagging with full-length fluorescent proteins (FP) can affect both the function and targeting of proteins. The strength of the system used in the current manuscript is that the shorter tag is beneficial for detection of a number of proteins whose targeting and/or function is affected by tagging with full length FPs.

Furthermore, the system used here can nicely detect mitochondrial pools of dually localized proteins. It is especially useful when these pools are minor and their signals are therefore easily masked by the strong signals coming from the major, nonmitochondrial pools of the proteins.

Weaknesses:

My only concern is that the biological significance of the screen performed appears limited. The dataset obtained is largely in agreement with several previous proteomic screens but it is, unfortunately, not more comprehensive than them, rather the opposite. For proteins that were identified inside mitochondria for the first time here or were identified in an unexpected location within the organelle, it remains unclear whether these localizations represent some minor, missorted pools of proteins or are indeed functionally important fractions and/or productive translocation intermediates. The authors also allude to several potential applications of the system but do little to explore any of these directions.

Comments on revisions:

The revised version of the manuscript submitted by Bykov et al addresses the comments and concerns raised by the Reviewers. It is a pity that the verification of the newly obtained data and its further biological exploration is apparently more challenging than perhaps anticipated.

---

## [Author Response]

The following is the authors’ response to the original reviews.

**Reviewer #1 (Public Review):**
Summary:The study conducted by the Schuldiner's group advances the understanding of mitochondrial biology through the utilization of their bi-genomic (BiG) split-GFP assay, which they had previously developed and reported. This research endeavors to consolidate the catalog of matrix and inner membrane mitochondrial proteins. In their approach, a genetic framework was employed wherein a GFP fragment (GFP1-10) is encoded within the mitochondrial genome. Subsequently, a collection of strains was created, with each strain expressing a distinct protein tagged with the GFP11 fragment. The reconstitution of GFP fluorescence occurs upon the import of the protein under examination into the mitochondria.

We are grateful for the positive evaluation. We would like to clarify that the bi-genomic (BiG) split-GFP assay was developed by the labs of H. Becker and Roza Kucharzyk by highly laborious construction of the strain with mtDNA-encoded GFP_1-10_ (Bader et al, 2020).

Strengths:Notably, this assay was executed under six distinct conditions, facilitating the visualization of approximately 400 mitochondrial proteins. Remarkably, 50 proteins were conclusively assigned to mitochondria for the first time through this methodology. The strains developed and the extensive dataset generated in this study serve as a valuable resource for the comprehensive study of mitochondrial biology. Specifically, it provides a list of 50 "eclipsed" proteins whose role in mitochondria remains to be characterized.Weaknesses:The work could include some functional studies of at least one of the newly identified 50 proteins.

In response to this we have expanded the characterization of phenotypic effects resulting from changing the targeting signal and expression levels of the dually localized Gpp1 protein and expanded the data in Fig. 3, panels H and I.

**Reviewer #2 (Public Review):**
The authors addressed the question of how mitochondrial proteins that are dually localized or only to a minor fraction localized to mitochondria can be visualized on the whole genome scale. For this, they used an established and previously published method called BiG split-GFP, in which GFP strands 1-10 are encoded in the mitochondrial DNA and fused the GFP11 strand C-terminally to the yeast ORFs using the C-SWAT library. The generated library was imaged under different growth and stress conditions and yielded positive mitochondrial localization for approximately 400 proteins. The strength of this method is the detection of proteins that are dually localized with only a minor fraction within mitochondria, which so far has hampered their visualization due to strong fluorescent signals from other cellular localizations. The weakness of this method is that due to the localization of the GFP1-10 in the mitochondrial matrix, only matrix proteins and IM proteins with their C-termini facing the matrix can be detected. Also, proteins that are assembled into multimeric complexes (which will be the case for probably a high number of matrix and inner membrane-localized proteins) resulting in the C-terminal GFP11 being buried are likely not detected as positive hits in this approach. Taking these limitations into consideration, the authors provide a new library that can help in the identification of eclipsed protein distribution within mitochondria, thus further increasing our knowledge of the complete mitochondrial proteome. The approach of global tagging of the yeast genome is the logical consequence after the successful establishment of the BiG split-GFP for mitochondria. The authors also propose that their approach can be applied to investigate the topology of inner membrane proteins, however, for this, the inherent issue remains that it cannot be excluded that even the small GFP11 tag can impact on protein biogenesis and topology. Thus, the approach will not overcome the need to assess protein topology analysis via biochemical approaches on endogenous untagged proteins.
**Reviewer #3 (Public Review):**
Summary:Here, Bykov et al move the bi-genomic split-GFP system they previously established to the genomewide level in order to obtain a more comprehensive list of mitochondrial matrix and inner membrane proteins. In this very elegant split-GFP system, the longer GFP fragment, GFP1-10, is encoded in the mitochondrial genome and the shorter one, GFP11, is C-terminally attached to every protein encoded in the genome of yeast *Saccharomyces cerevisiae*. GFP fluorescence can therefore only be reconstituted if the C-terminus of the protein is present in the mitochondrial matrix, either as part of a soluble protein, a peripheral membrane protein, or an integral inner membrane protein. The system, combined with high-throughput fluorescence microscopy of yeast cells grown under six different conditions, enabled the authors to visualize ca. 400 mitochondrial proteins, 50 of which were not visualised before and 8 of which were not shown to be mitochondrial before. The system appears to be particularly well suited for analysis of dually localized proteins and could potentially be used to study sorting pathways of mitochondrial inner membrane proteins.Strengths:Many fluorescence-based genome-wide screens were previously performed in yeast and were central to revealing the subcellular location of a large fraction of yeast proteome. Nonetheless, these screens also showed that tagging with full-length fluorescent proteins (FP) can affect both the function and targeting of proteins. The strength of the system used in the current manuscript is that the shorter tag is beneficial for the detection of a number of proteins whose targeting and/or function is affected by tagging with full-length FPs.Furthermore, the system used here can nicely detect mitochondrial pools of dually localized proteins. It is especially useful when these pools are minor and their signals are therefore easily masked by the strong signals coming from the major, nonmitochondrial pools of the proteins.Weaknesses:My only concern is that the biological significance of the screen performed appears limited. The dataset obtained is largely in agreement with several previous proteomic screens but it is, unfortunately, not more comprehensive than them, rather the opposite. For proteins that were identified inside mitochondria for the first time here or were identified in an unexpected location within the organelle, it remains unclear whether these localizations represent some minor, missorted pools of proteins or are indeed functionally important fractions and/or productive translocation intermediates. The authors also allude to several potential applications of the system but do little to explore any of these directions.

We agree with the reviewer that a single method may not be used for the construction of the complete protein inventory of an organelle or its sub-compartment. We suggest that the value of our assay is in providing a complementary view to the existing data and approaches. For example, we confirm the matrix localization of several proteins that were only found in the two proteomic data and never verified before (Vögtle et al, 2017; Morgenstern et al, 2017). Given that proteomics is a very sensitive technique and false positives are hard to completely exclude, our complementary verification is valuable.

**Reviewer #1 (Recommendations for the authors):**
In my opinion, the manuscript can be published as it is, and I would expect that future work will advance the functional properties of the newly found mitochondrial proteins.

We thank the reviewer for their positive evaluation

**Reviewer #2 (Recommendations for the authors)**
(1) Due to the localization of the GFP1-10 in the matrix, only matrix and IM proteins with C-termini facing the matrix can be detected, this should be added e.g. in the heading of the first results part and discussed earlier in the manuscript. In addition, the limitation that assembly into protein complexes will likely preclude detection of matrix and IM proteins needs to be discussed.

To address the first point, we edited the title of the first section to only mention the visualization of the matrix-facing proteome and remove the words “inner membrane”. We also clarified early in the Results section that we only consider the matrix-facing C-termini by extending the sentence early in the results section “To compare our findings with published data, we created a unified list of 395 proteins that are observed with high confidence using our assay indicating that their C-terminus is positioned in the matrix (Fig. 2 – figure supplement 1B-D, Table S1).” (P. 6 Lines 1-3). Concluding the comparison with the earlier proteomic studies we also added the sentence “Many proteins are missing because their C-termini are facing the IMS” (P.8 Line 2).

To address the second point concerning the possible interference of the complex assembly and protein detection by our assay, we conducted an additional analysis. The analysis takes advantage of the protein complexes with known structures where we could estimate if the C-terminus with the GFP_11_ tag would be available for GFP1-10 binding. We added the additional figure (Figure 3 – figure supplement 2) and following text in the Results section (P.7 Lines 22-34):

“To examine the influence of protein complex assembly on the performance of the BiG Mito-Split assay we analyzed the published structures of the mitoribosome and ATP synthase (Desai et al, 2017; Srivastava et al, 2018; Guo et al, 2017) and classified all proteins as either having C-termini in, or out of, the complex. There was no difference between the “in” and “out” groups in the percentage observed in the BiG Mito-Split collection (Fig. 3 – figure supplement 2A) suggesting that the majority of the GFP11tagged proteins have a chance to interact with GFP1-10 before (or instead of) assembling into the complex. PCR and western blot verification of eight strains with the tagged complex subunits for which we observed no signal showed that mitoribosomal proteins were incorrectly tagged or not expressed, and the ATP synthase subunits Atp7, Atp19, and Atp20 were expressed (Fig. 3 – Supplement 2B). Atp19 and Atp20 have their C-termini most likely oriented towards the IMS (Guo et al, 2017) while Atp7 is completely in the matrix and may be the one example of a subunit whose assembly into a complex prevents its detection by the BiG Mito-Split assay.”

We also consider related points on the interference of the tag and the influence of protein essentiality in the replies to points (3) and (12) of these reviews.

(2) The imaging data is of high quality, but the manuscript would greatly benefit from additional analysis to support the claims or hypothesis brought forward by the authors. The idea that the nonmitochondrial proteins are imported due to their high sequence similarity to MTS could be easily addressed at least for some of these proteins via import studies, as also suggested by the authors.

The idea that non-mitochondrial proteins may be imported into mitochondria due to occasional sequence similarity was recently demonstrated experimentally by (Oborská-Oplová et al, 2025). We incorporate this information in the Discussion section as follows (P. 14 Lines 10-16):

“It was also recently shown that the r-protein uS5 (encoded by RPS2 in yeast) has a latent MTS that is masked by a special mitochondrial avoidance segment (MAS) preceding it (Oborská-Oplová et al, 2025). The removal of the MAS leads to import of uS5 into mitochondria killing the cells. The case of uS5 is an example of occasional similarity between an r-protein and an MTS caused by similar requirements of positive charges for rRNA binding and mitochondrial import. It remains unclear if other r-proteins have a MAS and if there are other mechanisms that protect mitochondria from translocation of cytosolic proteins.”

We also conducted additional analysis to substantiate the claim that ribosomal (r)-proteins are similar in their physico-chemical properties to MTS-containing mitochondrial proteins. For this we chose not to use prediction algorithms like TartgetP and MitoFates that were already trained on the same dataset of yeast proteins to discriminate cytosolic and mitochondrial localization. Instead, we extended the analysis earlier made by (Woellhaf et al, 2014) and calculated several different properties such as charge, hydrophobicity, hydrophobic moment and amino acid content for mitochondrial MTS-containing proteins, cytosolic non-ribosomal proteins, and r-proteins. The analysis showed striking similarity of r-proteins and mitochondrial proteins. We incorporate a new Figure 3 – figure supplement 3 and the following text in the Results section (P. 8 Lines14-22):

“Five out of eight proteins are components of the cytosolic ribosome (r-proteins). In agreement with previous reports (Woellhaf et al, 2014) we find that their unique properties, such as charge, hydrophobicity and amino acid content, are indeed more similar to mitochondrial proteins than to cytosolic ones (Fig. 3 – figure supplement 3). Additional experiments with heterologous protein expression and in vitro import will be required to confirm the mitochondrial import and targeting mechanisms of these eight non-mitochondrial proteins. The data highlights that out of hundreds of very abundant proteins with high prediction scores only few are actually imported and highlights the importance of the mechanisms that help to avoid translocation of wrong proteins (Oborská-Oplová et al, 2025).”

To further prove the possibility of r-protein import into mitochondria we aimed to clone the r-proteins identified in this work for cell-free expression and import into purified mitochondria. Despite the large effort, we have succeeded in cloning and efficiently expressing only Rpl23a (Author response image 1 A). Rpl23a indeed forms proteinase-protected fractions in a membrane potential-dependent manner when incubated with mitochondria. The inverse import dynamics of Rpl23a could be either indicative of quick degradation inside mitochondria or of background signal during the import experiments (Author response image 1.A). To address the r-protein degradation possibility, we measured how does GFP signal change in the BiG Mito-Split diploid collection strains after blocking cytosolic translation with cycloheximide (CHX). For this we selected Mrpl12a, that had one of the highest signals. We did not detect any drop in fluorescence signal for Rpl12a and the control protein Mrpl6 (Author response image 1 B). This might indicate the lack of degradation, or the degradation of the whole protein except GFP_11_ that remains connected to GFP_1-10_. Due to time constrains we could not perform all experiments for the whole set of potentially imported r-proteins. Since more experiments are required to clearly show the mechanisms of mitochondrial r-protein import, degradation, and toxicity, or possible moonlighting functions (such as import into mitochondria derived from pim1∆ strain, degradation assays, fractionations, and analyses with antibodies for native proteins) we decided not to include this new data into the manuscript itself.

**Author response image 1. sa4fig1:** The import of r-proteins into mitochondria and their stability. (A) Rpl23 was synthesized in vitro (Input), radiolabeled, and imported into mitochondria isolated from BY4741 strain as described before (Peleh et al, 2015); the import was performed for 5,10, or 15 min and mitochondria were treated with proteinase K (PK) to degrade nonimported proteins; some reactions were treated with the mix of valinomycin, antimycin, and oligomycin (VAO) to dissipate mitochondrial membrane potential; the proteins were visualized by SDS-PAGE and autoradiography (B) The strains from the diploid BiG Mito-Split collection were grown in YPD to mid-logarithmic growth phase, then CHX was added to block translation and cell aliquots were taken from the culture and analyzed by fluorescence microscopy at the indicated time points. Scale bar is 5 µm.

(3) The claim that the approach can be used to assess the topology of inner membrane proteins is problematic as the C-terminal tag can alter the biogenesis pathway of the protein or impact on the translocation dynamics (in particular as the imaging method applied here does not allow for analysis of dynamics). The hypothesis that the biogenesis route can be monitored is therefore far-reaching. To strengthen the hypothesis the authors should assess if the C-terminal GFP11 influences protein solubility by assessing protein aggregation of e.g. Rip1.

We agree with the reviewer that the tag and assembly of GFP_1-10/11_ can further complicate the assessment of topology of the IM proteins that already have complex biogenesis routes (lateral transfer, conservative, and a Rip1-specific Bcs1 pathway). To emphasize that the assessment of the steady state topology needs to be backed up by additional biochemical approaches, we edited the beginning of the corresponding Results sections as follows (P.11 Lines 2-6):

“Studying membrane protein biogenesis requires an accurate way to determine topology in vivo. The mitochondrial IM is one of the most protein-rich membranes in the cell supporting a wide variety of TMD topologies with complex biogenesis pathways. We aimed to find out if our BiG Mito-Split collection can accurately visualize the steady-state localization of membrane protein C-termini protruding into the matrix or trap protein transport intermediates” (inserted text is underlined).

The collection that we studied by microscopy is diploid and contains one WT copy of each 3xGFP_11_tagged gene. To assess the influence of the tag on the protein function we performed growth assays with haploid strains which have one 3xGFP_11_-tagged gene copy and no GFP_1-10_. We find that Rip13xGFP_11_ displays slower growth on glycerol at 30°C and even slower at 37°C while tagged Qcr8, Qcr9, and Qcr10 grow normally (Author response image 2 A). Based on the growth assays and microscopy it is not possible to conclude whether the “Qcr” proteins’ biogenesis is affected by the tag. It may be that laterally sorted proteins are functional with the tag and constitute the majority while only a small portion is translocated into the matrix, trapped and visualized with GFP_1-10_. In case of Rip1 it was shown that C-terminal tag can affect its interaction with the chaperone Mzm1 and promote Rip1 aggregation (Cui et al, 2012). The extent of Rip1 function disruption can be different and depends on the tag. We hypothesize that our split-assay may trap the pre-translocation intermediate of Rip1 and can be helpful to study its interactors. To test this, we performed anti-GFP immune-precipitation (IP) using GFP-Trap beads (Author response image 2 B).

**Author response image 2. sa4fig2:** The influence of 3x-GFP11 on the function and processing of the inner membrane proteins. (A) Drop dilution assays with haploid strains from C-SWAT 3xGFP_11_ library on fermentative (YPD) and respiratory (YPGlycerol) media at different temperatures. (B) Immuno-precipitation with GFP-Trap agarose was performed on haploid strain that has only Rip1-3xGFP_11_ and on the diploid strain derived from this haploid mated with BiG Mito-Split strain containing mtGFP_1-10_ and WT untagged Rip1 using the lysis (1% TX-100) and washing protocols provided by the manufacturer; the total (T) and eluted with the Laemmli buffer (IP) samples were analyzed by immunoblotting with polyclonal rabbit antibodies against GFP (only visualizes GFP_11_ in these samples) and Rip1 (visualizes both tagged and WT Rip1). Polyclonal home-made rabbit antisera for GFP and Rip1 were kindly provided by Johannes Herrmann (Kaiserslautern) and Thomas Becker (Bonn); the antisera were diluted 1:500 for decorating the membranes.

We find that the haploid strain with Rip1-3xGFP_11_ contains not only mature (m) and intermediate (i) forms but also an additional higher Mw band that we interpreted as precursor that was not cleaved by MPP. WT Rip1 in the diploid added two more lower Mw bands: (m) and (i) forms of the untagged Rip1. IP successfully enriched GFP_1-10_ fragment as visualized by anti-GFP staining. Interestingly only the highest Mw Rip1-3xGFP_11_ band was also enriched when anti-Rip1 antibodies were used to analyze the samples. This suggests that Rip1 precursor gets completely imported and interacts with GFP_1-10_ and can be pulled down. It is however not processed. Processed Rip1 is not interacting with GFP_1-10_. Based on the literature we expect all Rip1 in the matrix to be cleaved by MPP including the one interacting with GFP. Due to this discrepancy, we did not include this data in the manuscript. This is however clear that the assay may be useful to analyze biogenesis intermediates of the IM and matrix proteins. To emphasize this, we added information on the C-terminal tagging of Rip1 in the Results section (P. 11 Lines 18-20):

“It was shown that a C-terminal tag on Rip1 can prevent its interaction with the chaperone Mzm1 and promote aggregation in the matrix (Cui et al, 2012). It is also possible that our assay visualizes this trapped biogenesis intermediate.”

We also added a note on biogenesis intermediates in the Discussion (P. 14 Line 36 onwards):

“It is possible that the proteins with C-termini that are translocated into the IMS from the matrix side can be trapped by the interaction with GFP_1-10_. In that case, our assay can be a useful tool to study these pre-translocation intermediates.”

(4) The hypothesis that the method can reveal new substrates for Bcs1 is interesting, and it would strongly increase the relevance for the scientific community if this would be directly tested, e.g. by deleting BCS1 and testing if more IM proteins are then detected by interaction with the matrix GFP110.

we attempted to move the BiG Mito-Split assay into haploid strains where BCS1 and other factors can be deleted, however, this was not successful. Since this was a big effort (We cloned 10 potential substrate proteins but none of them were expressed) we decided not to pursue this further.

(5) The screening of six different growth conditions reflects the strength of the high-throughput imaging readout. However, the interpretation of the data and additional follow-up on this is rather short and would be a nice addition to the present manuscript. In addition, one wonders, what was the rationale behind these six conditions (e.g. DTT treatment)? The direct metabolic shift from fermentation to respiration to boost mitochondrial biogenesis would be a highly interesting condition and the authors should consider adding this in the present manuscript.

we agree with the reviewer that the analysis of different conditions is a strength of this work. However, we did not reveal any clear protein groups with strong conditional import and thus it was hard to select a follow-up candidate. The selection of conditions was partially driven by the technical possibilities: the media change is challenging on the robotic system; heat shock conditions make microscope autofocus unstable; library strain growth on synthetic respiratory media is very slow and the media cannot be substituted with rich media due to its autofluorescence. However, the usage of the spinning disc confocal microscope allowed us to screen directly in synthetic oleate media which has a lot of background on widefield systems due to oil micelles. We extended the explanation of condition choice as follows (P. 4 Line 34 onwards):

“The diploid BiG Mito-Split collection was imaged in six conditions representing various carbon sources and a diversity of stressors the cells can adapt to: logarithmic growth on glucose as a control carbon source and oleic acid as a poorly studied carbon source; post-diauxic (stationary) phase after growth on glucose where mitochondria, are more active and inorganic phosphate (Pi) depletion that was recently described to enhance mitochondrial membrane potential (Ouyang et al, 2024); as stress conditions we chose growth on glucose in the presence of 1 mM dithiothreitol (DTT) that might interfere with the disulfide relay system in the IMS, and nitrogen starvation as a condition that may boost biosynthetic functions of mitochondria. DTT and nitrogen starvation were earlier used for a screen with the regular C’-GFP collection (Breker et al, 2013). Another important consideration for selecting the conditions was the technical feasibility to implement them on automated screening setups.”

**Reviewer #3 (Recommendations for the authors)**
(6) This is a very elegant and clearly written study. As mentioned above, my only concern is that the biological significance of the obtained data, at this stage, is rather limited. It would have been nice if the authors explored one of the potential applications of the system they propose. For example, it should be relatively easy to analyze whether Cox26, Qcr8, Qcr9, or Qcr10 are new substrates of Bsc1, as the authors speculate.

we thank the reviewer for their positive feedback. We addressed the biological application of the screen by including new data on metabolite concentrations in the strains where Gpp1 N-terminus was mutated leading to loss of the mitochondrial form. We added panels H and I to Figure 4, the new Supplementary Table S2 and appended the description of these results at the end of the third Results subsection (P. 10 Lines 19-35). Our data now show a role for the mitochondrial fraction of Gpp1 which adds mechanistic insight into this dually localized protein.

We also were interested in the applications of our system to the study of mitochondrial import. However, the study of Cox26, Qcr8, Qcr9, and Qcr10 was not successful (also related to point 4, Reviewer #2). We thus decided to investigate the import mechanisms of the poorly studied dually localized proteins Arc1, Fol3, and Hom6 (related to Figure 4 of the original manuscript). To this end, we expressed these proteins in vitro, radiolabeled, and performed import assays with purified mitochondria. Arc1 was not imported, Fol3 and Hom6 gave inconclusive results (Author response image 3). Since it is known that even some genuine fully or dually localized mitochondrial proteins such as Fum1 cannot be imported in vitro post-translationally (Knox et al, 1998), we cannot draw conclusions from these experiments and left them out of the revised manuscript. Additional investigation is required to clarify if there exist special cytosolic mechanisms for the import of these proteins that were not reconstituted in vitro such as co-translational import.

**Author response image 3. sa4fig3:** In vitro import of poorly studies dually localized proteins. Arc1, Fol3, and Hom6 were cloned into pGEM4 plasmid, synthesized in vitro (Input), radiolabeled, and imported into mitochondria isolated from BY4741 strain as described before (Peleh et al, 2015); the import was performed for 5,10, or 15 min and mitochondria were treated with proteinase K (PK) to degrade non-imported proteins; some reactions were treated with the mix of valinomycin, antimycin, and oligomycin (VAO) to dissipate mitochondrial membrane potential. The proteins were separated by SDS-PAGE and visualized by autoradiography.

Minor comments:(7) It is unclear why the authors used the six growth conditions they used, and why for example a nonfermentable medium was not included at all.

we address this shortcoming in the reply to the previous point 5 (Reviewer #2).

(8) Page 2, line 17 - "Its" should be corrected to "its".

Changed

(9) Page 2, line 25 to the end of the paragraph - the authors refer to the TIM complex when actually the TIM23 complex is probably meant. Also, it would be clearer if the TIM22 complex was introduced as well, especially in the context of the sentence stating that "the IM is a major protein delivery destination in mitochondria".

This was corrected.

(10) Page 5, line 35 - "who´s" should be corrected to "whose".

This was corrected.

(11) Page 9, line 5 - "," after Gpp1 should probably be "and".

This was corrected.

(12) Page 11 - the authors discuss in several places the possible effects of tags and how they may interfere with "expression, stability and targeting of proteins". Protein function may also be dramatically affected by tags - a quick look into the dataset shows that several mitochondrial matrix and inner membrane proteins that are essential for cell viability were not identified in the screen, likely because their function is impaired.

we agree with the reviewer that the influence of tags needs to be carefully evaluated. This is not always possible in the context of whole genomic screens. Sometimes, yeast collections (and proteomic datasets) can miss well-known mitochondrial residents without a clear reason. To address this important point we conducted an additional analysis to look specifically at the essential proteins. We indeed found that several of the mitochondrial proteins that are essential for viability were absent from the collection at the start, but for those present, their essentiality did not impact the likelihood to be detected in our assay. To describe the analysis we added the following text and a Fig. 3 – figure supplement 2. Results now read (P.7 Lines 8-21):

“Next, we checked the two categories of proteins likely to give biased results in high-throughput screens of tagged collections: proteins essential for viability, and molecular complex subunits. To look at the first category we split the proteomic dataset of soluble matrix proteins (Vögtle et al. 2017) into essential and non-essential ones according to the annotations in the Saccharomyces Genome Database (SGD) (Wong et al, 2023). We found that there was no significant difference in the proportion of detected proteins in both groups (17 and 20 % accordingly), despite essential proteins being less represented in the initial library (Fig. 3 – figure supplement 2A). From the three essential proteins of the (Vögtle et al. 2017) dataset for which the strains present in our library but showed no signal, two were nucleoporins Nup57 and Nup116, and one was a genuine mitochondrial protein Ssc1. Polymerase chain reaction (PCR) and western blot verification showed that the Ssc1 strain was incorrect (Fig. 3 – figure supplement 2B). We conclude that essential proteins are more likely to be absent or improperly tagged in the original C’-SWAT collection, but the essentiality does not affect the results of the BiG Mito-Split assay.”

Discussion (P. 13 Lines 23-26):

“We did not find that protein complex components or essential proteins are more likely to be falsenegatives. However, some essential proteins were absent from the collection to start with (Fig. 3 – figure supplement 2A). Thus, a small tag allows visualization of even complex proteins.”

From our data it is difficult to estimate the effect of tagging on protein function. We also addressed the effect of tagging Rip1 as well as performed growth assays on the tagged small “Qcr proteins” in the reply to point 3 (Reviewer #2). It is also difficult to estimate the effect of GFP_1-10_ and _11_ complex assembly on protein function since the presence of functional, unassembled GFP_11_ tagged pool cannot be ruled out in our assay.

Other changes

Figure and table numbers changed after new data additions.

A sentence added in the abstract to highlight the additional experiments on Gpp1 function: “We use structure-function analysis to characterize the dually localized protein Gpp1, revealing an upstream start codon that generates a mitochondrial targeting signal and explore its unique function.”

The reference to the PCR verification (Fig. 3 – Supplement 2B) of correct tagging of Ycr102c was added to the Results section (P.8 Line 6), western blot verification added on.

Added the Key Resources Table at the beginning of the Methods section.

Small grammar edits, see tracked changes.

References:

Bader G, Enkler L, Araiso Y, Hemmerle M, Binko K, Baranowska E, De Craene J-O, Ruer-Laventie J, Pieters J, Tribouillard-Tanvier D, et al (2020) Assigning mitochondrial localization of dual localized proteins using a yeast Bi-Genomic Mitochondrial-Split-GFP. eLife 9: e56649

Cui T-Z, Smith PM, Fox JL, Khalimonchuk O & Winge DR (2012) Late-Stage Maturation of the Rieske Fe/S Protein: Mzm1 Stabilizes Rip1 but Does Not Facilitate Its Translocation by the AAA ATPase Bcs1. Mol Cell Biol 32: 4400–4409

Desai N, Brown A, Amunts A & Ramakrishnan V (2017) The structure of the yeast mitochondrial ribosome. Science 355: 528–531

Guo H, Bueler SA & Rubinstein JL (2017) Atomic model for the dimeric FO region of mitochondrial ATP synthase. Science 358: 936–940

Knox C, Sass E, Neupert W & Pines O (1998) Import into Mitochondria, Folding and Retrograde Movement of Fumarase in Yeast. J Biol Chem 273: 25587–25593

Morgenstern M, Stiller SB, Lübbert P, Peikert CD, Dannenmaier S, Drepper F, Weill U, Höß P, Feuerstein R, Gebert M, et al (2017) Definition of a High-Confidence Mitochondrial Proteome at Quantitative Scale. Cell Rep 19: 2836–2852

Oborská-Oplová M, Geiger AG, Michel E, Klingauf-Nerurkar P, Dennerlein S, Bykov YS, Amodeo S, Schneider A, Schuldiner M, Rehling P, et al (2025) An avoidance segment resolves a lethal nuclear–mitochondrial targeting conflict during ribosome assembly. Nat Cell Biol 27: 336–346

Peleh V, Ramesh A & Herrmann JM (2015) Import of Proteins into Isolated Yeast Mitochondria. In Membrane Trafficking: Second Edition, Tang BL (ed) pp 37–50. New York, NY: Springer

Srivastava AP, Luo M, Zhou W, Symersky J, Bai D, Chambers MG, Faraldo-Gómez JD, Liao M & Mueller DM (2018) High-resolution cryo-EM analysis of the yeast ATP synthase in a lipid membrane. Science 360: eaas9699

Vögtle F-N, Burkhart JM, Gonczarowska-Jorge H, Kücükköse C, Taskin AA, Kopczynski D, Ahrends R, Mossmann D, Sickmann A, Zahedi RP, et al (2017) Landscape of submitochondrial protein distribution. Nat Commun 8: 290

Woellhaf MW, Hansen KG, Garth C & Herrmann JM (2014) Import of ribosomal proteins into yeast mitochondria. Biochem Cell Biol 92: 489–498